# PRUNINGBENCH: A COMPREHENSIVE BENCHMARK OF STRUCTURAL PRUNING

## ABSTRACT

Structural pruning has emerged as a promising approach for producing more efficient models. Nevertheless, the community suffers from a lack of standardized benchmarks and metrics, leaving the progress in this area not fully comprehended. To fill this gap, we present the first comprehensive benchmark, termed ***PruningBench***, for structural pruning. PruningBench showcases the following three characteristics: 1) PruningBench employs a unified and consistent framework for evaluating the effectiveness of diverse structural pruning techniques; 2) PruningBench systematically evaluates 16 existing pruning methods, encompassing a wide array of models (*e.g.*, CNNs and ViTs) and tasks (*e.g.*, classification and detection); 3) PruningBench provides easily implementable interfaces to facilitate the implementation of future pruning methods, and enables the subsequent researchers to incorporate their work into our leaderboards. We will provide an online pruning platform for customizing pruning tasks and reproducing all results in this paper. Codes will also be made publicly available.

## 1 INTRODUCTION

Model compression is an essential pursuit in the domain of machine learning, motivated by the necessity to strike a balance between model accuracy and computational efficiency. Various approaches have been developed to create more efficient models, including pruning (Han et al., 2015), quantization (Rastegari et al., 2016), decomposition (Denton et al., 2014), and knowledge distillation (Hinton et al., 2015; Wang et al., 2023b). Among the multitude of compression paradigms, pruning has proven itself to be remarkably effective and practical (Ding et al., 2021; Gao et al., 2021; Liang et al., 2021; Lin et al., 2020; Park et al., 2020; Wang et al., 2020; 2021a; Yu et al., 2018; Xu et al.; Fang et al., 2023). The aim of network pruning is to eliminate redundant parameters of a network to produce sparse models and potentially speed up the inference. Mainstream pruning approaches can be categorized into *structuural pruning* and *unstruchaal pruning*. Unstructured pruning typically involves directly zeroing partial weights without modifying the network structure; whereas structured pruning methods, although some require specific hardware support, can physically remove grouped parameters from the network, they effectively compress the network size, thus getting a wider domain of applications in practice.

Despite the extensive research on structural pruning, the community still suffers from a lack of standardized benchmarks and metrics, leaving the progress in this area not fully comprehended (Blalock et al., 2020; Wang et al., 2023a). Table 1 provides the experimental settings used in some representative papers on network pruning, which unveils three pitfalls in structure pruning evaluations in the current literature:

**Pitfall 1: Limited comparisons with SOTA.** Many works (*e.g.,* Liu et al. (2017); Li et al. (2016); Park et al. (2020); Hu et al. (2016); Tan & Motani (2020)) limit their evaluations to a comparison between the original and pruned models, without benchmarking against state-of-the-art methodologies. Similarly, certain approaches (*e.g.,* Wen et al. (2016); Wang et al. (2019a); Lee et al. (2020); Ye et al. (2018); Huang & Wang (2018); Wen et al. (2016); Molchanov et al. (2019)) restrict their assessments to a single competitor. While some works endeavor to include more competitors, they exclusively compare themselves with a few methods within their specific subdomains (*e.g.,* norm-based, gradient-based) He et al. (2019); Wang et al. (2019b); He et al. (2017); Yu et al. (2018); Wang et al. (2021b); Sui et al. (2021); Zhang et al. (2021); Luo et al. (2017); Molchanov et al. (2016) rather

Table 1: Experimental settings in some representative structural pruning methods. "#Comp." indicates the number of pruning methods compared with the proposed method in the original paper. "Params" and "FLOPs" indicate whether parameter count and FLOPs are controlled when compared with alternative methods. "Regularizer" means whether a sparsity regularizer is employed.

| Methods | Pretrained Models | #Comp. | Pruning | Iteration | Params | FLOPs | Regul. |
|---|---|---|---|---|---|---|---|
| OBD-C (Wang et al., 2019a) | VGG19, ResNet32, PreResNet29 | 1 | local | once/iterative | × | × | × |
| Taylor (Molchanov et al., 2019) | LeNet, ResNet18 | 2 | global | iterative | × | × | × |
| FPGM (He et al., 2019) | ResNet18/20/32/34/50/56/101/110 | 3 | local | iterative | × | × | × |
| Magnitude (Li et al., 2016) | VGG16, ResNet34/56/110 | 0 | local | once/iterative | × | × | × |
| Random (Mittal et al., 2018) | VGG16, ResNet50 | 4 | local | once | × | × | × |
| LAMP (Lee et al., 2020) | VGG16, ResNet20/34, DenseNet121 | 4 | global | iterative | ✓ | × | × |
| HRank (Lin et al., 2020) | VGG16, GoogLeNet, ResNet56/110 | 4 | local | once | × | × | × |
| CP (He et al., 2017) | VGG16, ResNet50, Xception50 | 1 | local | once | × | ✓ | × |
| ThiNet (Luo et al., 2017) | VGG16, ResNet50 | 3 | local | once | × | × | × |
| NISP (Yu et al., 2018) | LeNet, AlexNet, GoogLeNet+once | 1 | global | once | × | × | × |
| BNScale (Liu et al., 2017) | VGGNet, DenseNet40, ResNet164 | 0 | global | once/iterative | × | × | ✓ |
| SSL (Wen et al., 2016) | LeNet, MLP, ConvNet, ResNet | 1 | local | once | × | × | ✓ |
| GrowingReg (Wang et al., 2020) | VGG19, ResNet56 | 5 | local | iterative | × | × | ✓ |

than conducting a broader comparison. Moreover, existing pruning methods are primarily tested on image classification tasks with CNNs, leaving their performance on other architectures or tasks largely unexplored.

**Pitfall 2: Inconsistent experimental settings.** Existing studies typically conduct evaluations under inconsistent experimental conditions, as illustrated in Table 1. For instance, previous works utilize varied pre-trained models for pruning. Different methodologies may employ distinct pruning techniques, such as local pruning (Wang et al., 2019a; He et al., 2019; Li et al., 2016; Mittal et al., 2018; Lin et al., 2020; He et al., 2017; Luo et al., 2017; Wen et al., 2016; Wang et al., 2020) and global pruning (Molchanov et al., 2019; Fang et al., 2023; Lee et al., 2020; Yu et al., 2018; Liu et al., 2017; Fang et al., 2023)). Furthermore, some approaches incorporate sparsity regularizers for pruning (You et al., 2019; Zhuang et al., 2020; Ye et al., 2018; Kang & Han, 2020; Li et al., 2020; Huang & Wang, 2018; Wen et al., 2016; Wang et al., 2020), yet compared with methods that do not integrate these regularizers. These inconsistent settings lead to biased performance comparisons and potentially misleading results.

**Pitfall 3: Comparisons without controlling variables.** Current methods usually present the changes in parameters (Park et al., 2020; Lee et al., 2020; Alizadeh et al., 2022; Gonzalez-Carabarin et al., 2022; Rachwan et al., 2022; Salehinejad & Valaee, 2021; Hu et al., 2016; Tan & Motani, 2020), FLOPs (Ding et al., 2021; Fang et al., 2023; Gao et al., 2021; He et al., 2017; Wang et al., 2021b; Zhang et al., 2021; Ding et al., 2019; Ye et al., 2020; Wang et al., 2020), or both (Wang et al., 2021a; Yu et al., 2018; Wang et al., 2019a; He et al., 2019; Lin et al., 2020; Molchanov et al., 2019; Yvinec et al., 2021; Kang & Han, 2020) after pruning, but neglect the consistency of these variables when comparing different methods. In different methods, the usage of parameters and FLOPS is inconsistent, which makes it impossible to compare the pruning results. Since accuracy, model size, and computational load all differ significantly after pruning, such comparisons without controlling variables can be hard to comprehend and leave the state of the field confusing.

To address aforementioned issues, we present to our best knowledge the first comprehensive benchmark, termed *PruningBench*, for structural pruning. In summary, the proposed PruningBench exhibits following three key characteristics.

(1) PruningBench employs a unified framework to evaluate existing diverse structural pruning techniques. Specially, PruningBench employs DepGraph Fang et al. (2023) to automatically group the network parameters, avoiding the labor effort and the group divergence by manually-designed grouping. Furthermore, PruningBench employs iterative pruning where a portion of parameters are removed per iteration until the controlled variable (*e.g.,* FLOPS) is reached. This standardized framework ensures more equitable and comprehensible comparisons among various pruning methods.

(2) PruningBench systematically evaluates 16 existing structural pruning methods, encompassing a wide array of models (ResNet18, ResNet50, VGG19, ViT (Dosovitskiy et al., 2020), YOLOv8 Jocher et al. (2023)) and tasks (*e.g.*, classification on CIFAR Krizhevsky et al. (2009) and ImageNet (Krizhevsky et al., 2017), detection on COCO (Lin et al., 2014)). In total, Pruning-Bench now has completed 645 model pruning experiments, yielding 13 leaderboards and a handful

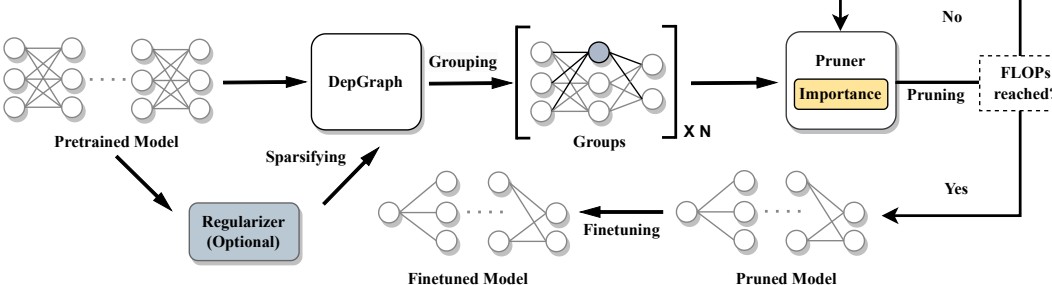

Figure 1: The framework of PruningBench, consisting of four steps: sparsifying, grouping, pruning and finetuning. Note that when benchmarking sparsifying regularizers (importance criteria), all other steps are fixed for fair comparisons.

of interesting findings which are not explored previously. We believe such a benchmark provides us with a more comprehensive picture of the state of the field, highlighting promising directions for future research.

(3) PruningBench is designed as an expandable package that standardizes experimental settings and eases the integration of new algorithmic implementations. PruningBench provides straightforward interfaces for implementing importance criteria methods and sparsity regularizers, facilitating the development, evaluation and integration of future pruning algorithms into the leaderboards (further details about the interfaces can be referred to in the A.3.1). Furthermore, our online platform enables users to customize pruning tasks by selecting models, datasets, methods, and hyperparameters, facilitating the reproducibility of the results presented in the paper.

**Reproducibility.** Leaderboards and online pruning platform will be available. Benchmarking more models is still in progress. The code will be made publicly available soon.

## 2 PRUNINGBENCH FRAMEWORK

**The PruningBench Framework.** The framework of the proposed PruningBench consists of four stages, as summarized in Figure 1. ❶ **Sparsifying**: Given a pretrained model to be compressed, PruningBench first employs a sparsity regularizer to sparsify model parameters. Note that this stage is skipped when benchmarking methods on importance criteria. ❷ **Grouping**: DepGraph Fang et al. (2023) is employed to model layer interdependencies and cluster coupled layers into groups. The following pruning is carried out at the group level. ❸ **Pruning**: PruningBench adopts iterative pruning to precisely control the model complexity of the pruned model to the predefined value. Before pruning, an importance criterion is selected for calculating the importance scores for the group parameters. Given a target pruning ratio $\alpha$, and the model is pruned by $S$ iterations. At each iteration, $\frac{\alpha}{S}$ of the parameters are pruned by thresholding the importance score. ❹ **Finetuning**: After pruning, PruningBench finetunes the pruned model, of which the accuracy is used for benchmark comparisons. Grouping stage and the finetuning stage are fixed the same for benchmarking all pruning methods.

Existing structural pruning literature mainly focuses on the sparsifying stage and the pruning stage. For sparsifying-stage methods, sparsity regularizers are proposed to learn structured sparse networks by imposing sparse constraints on loss functions and zeroing out certain weights. For pruning-stage methods, importance criteria are proposed to assess the importance of filters within a neural network, identifying redundant filters or channels that should be pruned. Note that importance criteria and sparsity regularizers are not mutually exclusive, suggesting that they can be utilized simultaneously to further promote the pruning performance. In this work, sparsifying-stage methods and pruning-stage methods are benchmarked separately.

## 3 PRUNINGBENCH SETTINGS

With the proposed PruningBench framework, we make a comprehensive study on existing structural pruning methods. We provide two leaderboards for each model-dataset combination, one for

sparsifying-stage methods and the other for pruning-stage methods. The experimental settings in our benchmark are summarized as follows. For more details, please refer to the A.2 .

**Models and Datasets.** The benchmark now has been conducted on visual classification and detection tasks. For visual classification, we carry out the pruning experiments on the widely used CIFAR100 Krizhevsky et al. (2009) and ImageNet Krizhevsky et al. (2017) datasets, with ResNet18, ResNet50 (He et al., 2016), VGG19 Simonyan & Zisserman (2014) and ViT-small (Dosovitskiy et al., 2020). For visual detection, evaluations are conducted with YOLOv8 Jocher et al. (2023) on the COCO dataset (Lin et al., 2014). The ResNet models for CIFAR100 are sourced from (Lab, 2023), and VGG models are sourced from (Tian, 2019). For all these CIFAR models, the pretrained models used for pruning are trained by ourselves (see A.2.2 for the training details). For ImageNet experiments, the ResNet models and pretrained weights are obtained from the torchvision library (Pytorch, 2023), while the ViT-small model and its pretrained weight are sourced from the timm library (Wightman, 2019). For COCO experiments, the implementation and pretrained weight of the YOLOv8 model are obtained from the ultralytics library (ultralytics, 2023).

**Pruning Methods.** As aforementioned, PruningBench systematically evaluates 16 existing pruning methods, including both the sparsifying-stage methods and the pruning-stage methods. For sparsifying-stage methods, we select GrowingReg (Wang et al., 2020), GroupNorm (Fang et al., 2023), GroupLASSO (Friedman et al., 2010), and BNScale (Liu et al., 2017), where GrowingReg, GroupNorm and GroupLASSO are representatives of weight-based sparsity regularizers, and BN-Scale is a BN-based sparsity regularizer. Pruning-stage methods can be further categorized into *data-free* and *data-driven* methods. Data-free methods rely solely on weight information and produce deterministic results, whereas data-driven methods require input samples for pruning and yield non-deterministic results. In our benchmark, we select MagnitudeL1 (Li et al., 2016), MagnitudeL2 (Li et al., 2016), LAMP (Lee et al., 2020), FPGM (He et al., 2019), Random (Mittal et al., 2018) and BNScale Liu et al. (2017) as the representatives of the data-free methods, and CP (He et al., 2017), HRank (Lin et al., 2020), ThiNet (Luo et al., 2017), OBD-C (Wang et al., 2019a), OBD-Hessian (Fang et al., 2023), and Taylor Molchanov et al. (2019) as the representatives of the data-driven methods. Notably, for data-driven methods, we observe varying results due to varying input samples. To mitigate this randomness, we repeat the experiments three times to get the average results.

**Performance Metrics.** The performances of different pruning methods are evaluated at several predefined speedup ratios.On CIFAR100, the speedup ratios are defined as {2x, 4x, 8x}. For large datasets, COCO and ImageNet, where tasks become more complex and tolerable pruning decreases, we adopt the speedup ratios {2x, 3x, 4x}. Note that speedup ratio represents the difference in FLOPS between the pruned model and the original model. At each speedup ratio, pruning methods are compared in metrics of accuracy, parameters, pruning time, and regularizing time if sparsity regularizers are used.

**Pruning Schemes.** Currently there exist two pruning schemes: *local pruning* and *global pruning*. Local pruning removes a consistent proportion of parameters for each group in the network (Wang et al., 2019a; He et al., 2019; Li et al., 2016; Mittal et al., 2018; Lin et al., 2020; He et al., 2017; Luo et al., 2017; Wen et al., 2016; Wang et al., 2020). However, as the importance and the redundancy of parameters across layers differ largely, a consistent pruning strategy is usually suboptimal. In contrast to local pruning, global pruning removes structures from all available structures of a network until a specific speedup ratio is reached (Molchanov et al., 2019; Fang et al., 2023; Lee et al., 2020; Yu et al., 2018; Liu et al., 2017; Fang et al., 2023), without constraining the pruning ratio across different groups to be consistent. However, global pruning may prune the entire group at a high speedup ratio, leaving the model functionality broken down. To address these issues, we propose protected global pruning in this benchmark , which preserves at least 10% of the parameters within each group with global pruning. Experiments demonstrate that protected global protection yields comparable results at low (*e.g.,* 2x) speedup ratio and significantly superior performance at high (*e.g.,* 4x and 8x) speedup ratio. Like other studies, we also adopt the pruning strategy that controls FLOPS. Experimental results and discussions are deferred to Section 4.

**Hyperparameters.** When evaluating pruning-stage methods (*i.e.,* importance criteria), the sparsifying stage is skipped. All involved hyperparameters in the fine-tuning stage are fixed to be the same. For sparsifying-stage methods, however, evaluation becomes more complex. Sparsifying-stage methods still rely on importance criteria at the pruning stage. For CNN experiments, we employ MagnitudeL2 Li et al. (2016) and BNScale Liu et al. (2017) when benchmarking sparsifying-stage

Table 2: The leaderboard of ResNet50 on CIFAR100 at three different speedup ratios, including rankings and the pruning results. "Step Time" indicates the time required for each pruning step, while "Reg Time" represents the time for each sparse learning epoch. An asterisk (*) indicates the importance criterion is random or data-driven that requires feature maps, gradients, *etc.,* to calculate importance, exhibiting stochastic behavior.

| Speed Up | Method | | Rank | Base | Pruned | Δ Acc | Pruning Ratio | Step Time | Reg Time |
|---|---|---|---|---|---|---|---|---|---|
| | Importance | Regularizer | | | | | | | |
| 2x | OBD-C* | N/A | 1 | 78.35 | 78.68 | +0.33 | 16.45 M (69.39%) | 7.559s | N/A |
| | Taylor* | N/A | 2 | 78.35 | 78.51 | +0.16 | 16.65 M (70.24%) | 3.740s | N/A |
| | FPGM | N/A | 3 | 78.35 | 78.37 | +0.02 | 15.37 M (64.84%) | 0.163s | N/A |
| | MagnitudeL2 | N/A | 4 | 78.35 | 78.32 | -0.03 | 16.63 M (70.17%) | 0.136s | N/A |
| | BNScale | N/A | 5 | 78.35 | 78.30 | -0.05 | 15.96 M (67.32%) | 0.141s | N/A |
| | ThiNet* | N/A | 6 | 78.35 | 78.14 | -0.21 | 15.19 M (64.06%) | 33.619s | N/A |
| | Random* | N/A | 7 | 78.35 | 77.97 | -0.38 | 11.78 M (49.70%) | 0.104s | N/A |
| | CP* | N/A | 8 | 78.35 | 77.80 | -0.55 | 7.15 M (30.15%) | 2m51s | N/A |
| | MagnitudeL1 | N/A | 9 | 78.35 | 77.62 | -0.73 | 16.91 M (71.34%) | 0.137s | N/A |
| | OBD-Hessian* | N/A | 10 | 78.35 | 77.26 | -1.09 | 7.83 M (33.03%) | 5m5s | N/A |
| | LAMP | N/A | 11 | 78.35 | 76.26 | -2.09 | 16.21 M (68.37%) | 0.150s | N/A |
| | HRank* | N/A | 12 | 78.35 | 76.13 | -2.22 | 6.47 M (27.29%) | 34m32s | N/A |
| | MagnitudeL2 | GroupLASSO | 1 | 78.35 | 78.73 | +0.38 | 16.51 M (69.66%) | 0.136s | 3m5s |
| | BNScale | BNScale | 2 | 78.35 | 78.36 | +0.01 | 15.97 M (67.37%) | 0.141s | 2m14s |
| | MagnitudeL2 | GroupNorm | 3 | 78.35 | 78.30 | -0.05 | 15.03 M (63.41%) | 0.136s | 3m7s |
| | BNScale | GroupLASSO | 4 | 78.35 | 78.24 | -0.11 | 15.86 M (66.90%) | 0.141s | 2m38s |
| | MagnitudeL2 | GrowingReg | 5 | 78.35 | 77.99 | -0.36 | 16.61 M (70.06%) | 0.136s | 3m1s |
| 4x | FPGM | N/A | 1 | 78.35 | 78.02 | -0.33 | 10.23 M (43.16%) | 0.163s | N/A |
| | MagnitudeL2 | N/A | 2 | 78.35 | 77.98 | -0.37 | 10.71 M (45.19%) | 0.136s | N/A |
| | BNScale | N/A | 3 | 78.35 | 77.90 | -0.45 | 10.53 M (44.41%) | 0.141s | N/A |
| | MagnitudeL1 | N/A | 4 | 78.35 | 77.82 | -0.53 | 11.10 M (46.81%) | 0.137s | N/A |
| | Taylor* | N/A | 5 | 78.35 | 77.69 | -0.66 | 5.47 M (23.09%) | 3.740s | N/A |
| | OBD-C* | N/A | 6 | 78.35 | 77.51 | -0.84 | 5.84 M (24.64%) | 7.559s | N/A |
| | Random* | N/A | 7 | 78.35 | 77.41 | -0.94 | 5.95 M (25.11%) | 0.104s | N/A |
| | ThiNet* | N/A | 8 | 78.35 | 77.23 | -1.12 | 4.72 M (19.91%) | 33.619s | N/A |
| | CP* | N/A | 9 | 78.35 | 75.68 | -2.67 | 2.65 M (11.18%) | 2m51s | N/A |
| | LAMP | N/A | 10 | 78.35 | 75.52 | -2.83 | 5.93 M (25.03%) | 0.150s | N/A |
| | OBD-Hessian* | N/A | 11 | 78.35 | 75.49 | -2.86 | 3.26 M (13.75%) | 5m5s | N/A |
| | HRank* | N/A | 12 | 78.35 | 73.76 | -4.59 | 1.69 M (7.11%) | 34m32s | N/A |
| | BNScale | BNScale | 1 | 78.35 | 78.16 | -0.19 | 10.37 M (43.75%) | 0.141s | 2m14s |
| | MagnitudeL2 | GroupLASSO | 2 | 78.35 | 78.01 | -0.34 | 10.79 M (45.53%) | 0.136s | 3m5s |
| | BNScale | GroupLASSO | 3 | 78.35 | 77.90 | -0.45 | 10.76 M (45.38%) | 0.141s | 2m38s |
| | MagnitudeL2 | GroupNorm | 4 | 78.35 | 77.88 | -0.47 | 9.84 M (41.51%) | 0.136s | 3m7s |
| | MagnitudeL2 | GrowingReg | 5 | 78.35 | 77.86 | -0.49 | 10.77 M (45.43%) | 0.136s | 3m1s |
| 8x | MagnitudeL1 | N/A | 1 | 78.35 | 76.99 | -1.36 | 6.82 M (28.77%) | 0.137s | N/A |
| | MagnitudeL2 | N/A | 2 | 78.35 | 76.38 | -1.97 | 6.89 M (29.05%) | 0.136s | N/A |
| | Random* | N/A | 3 | 78.35 | 76.13 | -2.22 | 2.98 M (12.57%) | 0.104s | N/A |
| | FPGM | N/A | 4 | 78.35 | 75.93 | -2.42 | 7.16 M (30.20%) | 0.163s | N/A |
| | BNScale | N/A | 5 | 78.35 | 75.81 | -2.54 | 6.69 M (28.22%) | 0.141s | N/A |
| | OBD-C* | N/A | 6 | 78.35 | 75.78 | -2.57 | 2.35 M (9.92%) | 7.559s | N/A |
| | Taylor* | N/A | 7 | 78.35 | 75.38 | -2.97 | 1.98 M (8.34%) | 3.740s | N/A |
| | ThiNet* | N/A | 8 | 78.35 | 75.29 | -3.06 | 1.58 M (6.68%) | 33.619s | N/A |
| | OBD-Hessian* | N/A | 9 | 78.35 | 74.49 | -3.86 | 1.66 M (7.02%) | 5m5s | N/A |
| | LAMP | N/A | 10 | 78.35 | 73.48 | -4.87 | 3.62 M (15.27%) | 0.150s | N/A |
| | CP* | N/A | 11 | 78.35 | 72.39 | -5.96 | 0.97 M (4.07%) | 2m51s | N/A |
| | HRank* | N/A | 12 | 78.35 | 70.54 | -7.81 | 0.64 M (2.69%) | 34m32s | N/A |
| | MagnitudeL2 | GrowingReg | 1 | 78.35 | 76.39 | -1.96 | 7.00 M (29.52%) | 0.136s | 3m1s |
| | MagnitudeL2 | GroupLASSO | 2 | 78.35 | 76.27 | -2.08 | 7.09 M (29.90%) | 0.136s | 3m5s |
| | MagnitudeL2 | GroupNorm | 3 | 78.35 | 75.93 | -2.42 | 7.18 M (30.28%) | 0.136s | 3m7s |
| | BNScale | GroupLASSO | 4 | 78.35 | 75.60 | -2.75 | 7.19 M (30.32%) | 0.141s | 2m38s |
| | BNScale | BNScale | 5 | 78.35 | 75.47 | -2.88 | 6.90 M (29.12%) | 0.141s | 2m14s |

methods, which are proven to be stable and data-agnostic. However, for ViT experiments, we only use MagnitudeL2 due to the incompatibility of the ViT architecture with BNScale. Moreover, different sparsity regularizers have different hyperparameters, which are specific to each case and exhibit substantial differences across diverse model-dataset tasks. In our experiments, we carefully tune the hyperparameters individually for each sparsity regularizer. For more hyperparameters of sparsifying, pruning and finetuning stages, please refer to the A.2.3 A.2.4.

## 4 BENCHMARK RESULTS AND DISCUSSIONS

PruningBench now has completed 645 model pruning experiments (*i.e.,* getting 645 pruned models), yielding 13 leaderboards (9 on CIFAR, 3 on ImageNet, and one on COCO). For space considerations, here we present the leaderboard results of ResNet50 on CIFAR100 in Table 2, and the leaderboard

Table 3: The leaderboard of ViT-small on ImageNet at three different speedup ratios.

| Speed Up | Method | | Rank | Base | Pruned | △ Acc | Parameters | Step Time | Reg Time |
|---|---|---|---|---|---|---|---|---|---|
| | Importance | Regularizer | | | | | | | |
| 2x | FPGM | N/A | 1 | 78.588 | 69.248 | -9.34 | 10.365 M (47.01%) | 0.937s | N/A |
| | Random* | N/A | 2 | 78.588 | 68.810 | -9.778 | 9.305 M (42.20%) | 0.888s | N/A |
| | LAMP | N/A | 3 | 78.588 | 68.724 | -9.864 | 10.169 M (46.12%) | 1.284s | N/A |
| | MagnitudeL1 | N/A | 4 | 78.588 | 68.602 | -9.986 | 10.375 M (47.05%) | 1.005s | N/A |
| | MagnitudeL2 | N/A | 5 | 78.588 | 68.316 | -10.272 | 10.346 M (46.92%) | 0.995s | N/A |
| | OBD-Hessian* | N/A | 6 | 78.588 | 67.514 | -11.074 | 10.334 M (46.87%) | 6m40s | N/A |
| | Taylor* | N/A | 7 | 78.588 | 67.400 | -11.188 | 10.468 M (47.47%) | 27.634s | N/A |
| | CP* | N/A | 7 | 78.588 | 67.400 | -11.188 | 10.334 M (46.87%) | 15m4s | N/A |
| | ThiNet* | N/A | 8 | 78.588 | 63.914 | -14.674 | 6.439 M (29.20%) | 3m17s | N/A |
| | MagnitudeL2 | GrowingReg | 1 | 78.588 | 68.715 | -9.873 | 10.359 M (46.98%) | 0.995s | 5h10m31s |
| | MagnitudeL2 | GroupNorm | 2 | 78.588 | 68.594 | -9.994 | 10.363 M (47.00%) | 0.995s | 5h21m21s |
| | MagnitudeL2 | GroupLASSO | 3 | 78.588 | 68.350 | -10.238 | 10.360 M (46.98%) | 0.995s | 5h15m13s |
| 3x | MagnitudeL1 | N/A | 1 | 78.588 | 63.120 | -15.468 | 6.57 M (29.79%) | 1.005s | N/A |
| | LAMP | N/A | 2 | 78.588 | 62.538 | -16.050 | 6.08 M (27.57%) | 1.284s | N/A |
| | MagnitudeL2 | N/A | 3 | 78.588 | 62.342 | -16.246 | 6.37 M (28.89%) | 0.995s | N/A |
| | Taylor* | N/A | 4 | 78.588 | 61.582 | -17.006 | 6.62 M (30.01%) | 27.634s | N/A |
| | FPGM | N/A | 5 | 78.588 | 60.660 | -17.928 | 5.701 M (25.85%) | 0.937s | N/A |
| | CP* | N/A | 6 | 78.588 | 56.626 | -21.962 | 6.778 M (30.74%) | 15m4s | N/A |
| | OBD-Hessian* | N/A | 7 | 78.588 | 54.796 | -23.792 | 6.39 M (28.98%) | 6m40s | N/A |
| | ThiNet* | N/A | 8 | 78.588 | 49.654 | -28.934 | 5.113 M (23.19%) | 3m17s | N/A |
| | Random* | N/A | 9 | 78.588 | 44.654 | -33.954 | 4.95 M (22.45%) | 0.888s | N/A |
| | MagnitudeL2 | GrowingReg | 1 | 78.588 | 62.608 | -15.980 | 6.57 M (29.81%) | 0.995s | 5h10m31s |
| | MagnitudeL2 | GroupNorm | 2 | 78.588 | 61.716 | -16.872 | 6.88 M (31.20%) | 0.995s | 5h21m21s |
| | MagnitudeL2 | GroupLASSO | 3 | 78.588 | 61.340 | -17.248 | 6.57 M (29.13%) | 0.995s | 5h15m13s |
| 4x | MagnitudeL1 | N/A | 1 | 78.588 | 59.950 | -18.638 | 5.06 M (22.93%) | 1.005s | N/A |
| | MagnitudeL2 | N/A | 2 | 78.588 | 59.082 | -19.506 | 4.89 M (22.15%) | 0.995s | N/A |
| | Taylor* | N/A | 3 | 78.588 | 57.650 | -20.938 | 4.80 M (21.76%) | 27.634s | N/A |
| | LAMP | N/A | 4 | 78.588 | 55.750 | -22.838 | 4.32 M (19.57%) | 1.284s | N/A |
| | FPGM | N/A | 5 | 78.588 | 48.258 | -30.33 | 3.25 M (14.74%) | 0.937 | N/A |
| | OBD-Hessian* | N/A | 6 | 78.588 | 36.600 | -41.988 | 4.25 M (19.27%) | 6m40s | N/A |
| | CP* | N/A | 7 | 78.588 | 42.574 | -36.014 | 5.253 M (23.82%) | 15m4s | N/A |
| | ThiNet* | N/A | 8 | 78.588 | 28.422 | -50.166 | 2.669 M (12.10%) | 3m17s | N/A |
| | Random* | N/A | 9 | 78.588 | 27.722 | -50.866 | 2.76 M (12.54%) | 0.888s | N/A |
| | MagnitudeL2 | GrowingReg | 1 | 78.588 | 59.630 | -18.958 | 4.56 M (20.66%) | 0.995s | 5h10m31s |
| | MagnitudeL2 | GroupLASSO | 2 | 78.588 | 57.312 | -21.276 | 4.59 M (20.81%) | 0.995s | 5h15m13s |
| | MagnitudeL2 | GroupNorm | 3 | 78.588 | 56.446 | -22.142 | 4.77 M (21.62%) | 0.995s | 5h21m21s |

results of ViT-small on ImageNet in Table 3. For more leaderboards, please refer to Table 9 to 21 in A.5.

## 4.1 BENCHMARK RESULTS

**Overall Results.** In general, no single method consistently outperforms the others across all settings and tasks. Nonetheless, weight norm-based methods, such as MagnitudeL1 and MagnitudeL2, typically exhibit superior performance and yield more reliable results, ranking within the top 5 in most rankings while maintaining computational efficiency. This is followed by BNScale, FPGM, Taylor, and OBD-C, which also show commendable results in various scenarios. Other methodologies may not exhibit significant overall advantages, but may perform well in specific situations. Now we provide more detailed analyses of the leaderboard results by answering the following questions.

### *Q1: What is the impact of the model architectures on the leaderboard rankings?*

**Observation:** BNScale, Hrank, and LAMP demonstrate clear architectural preferences. BNScale consistently ranks within the top five in most rankings for CNNs utilizing residual blocks (such as ResNet18, ResNet50, and YOLOv8), yet its efficacy notably diminishes when applied to VGG, where it typically ranks between 7th and 9th. In contrast, LAMP and Hrank display subpar performance on ResNet models, but showcase excellence on VGG, frequently ranking within the top 5. LAMP also demonstrates robust performance on ViT and YOLO, often ranking between 1st and 4th. While other pruning techniques exhibit some variability in performance across diverse architectures, they do not manifest strong architectural preferences.

### *Q2: What is the impact of the speedup ratio on the leaderboard rankings?*

**Observation:** Different methods can exhibit varying rankings under different speedup ratios. MagnitudeL1, MagnitudeL2, BNScale, and LAMP slightly improve in ranking as the speedup ratio increases, indicating a certain level of pruning resilience. Conversely, FPGM, ThiNet, and Hrank tend

to experience a decline in rankings as the speedup ratio increases. OBD-Hessian and CP methods show relatively stable performance, with minimal ranking shifts across speedup ratios. Taylor and OBD-C, however, display more erratic behavior, with their rankings sometimes rising or falling significantly depending on the architecture and speedup ratios.

### Q3: Which methods are more efficient in terms of computation time?

**Observation:** Obviously, sparsifying-stage methods are significantly more computation expensive than pruning-stage methods due to the cumbersome sparse learning process. In general, in our experiments sparsifying-stage methods take about $1.33 \sim 2$ times longer time than pruning-stage methods. For pruning-stage methods, data-driven importance criteria (Yu et al., 2018; He et al., 2017; Wang et al., 2019a; Lin et al., 2020; Molchanov et al., 2019; Luo et al., 2017; Mittal et al., 2018; LeCun et al., 1989; Fang et al., 2023), particularly those involving non-parallel operations (Lin et al., 2020; Fang et al., 2023; He et al., 2017; Luo et al., 2017), consume longer pruning time compared to data-free methods. For example, OBD-Hessian Fang et al. (2023) computes gradients separately for each sample. Thinet (Luo et al., 2017) and HRank Lin et al. (2020) determine the importance of each output channel of each layer individually. These techniques are well-suited for one-shot pruning, where importance scores are calculated only once, followed by pruning the network to achieve the target pruning ratio. However, when applied to iterative pruning, the computation time increases significantly as importance scores need to be calculated every iteration.

### Q4: How do sparsity regularizers improve the performance of prunned model?

**Observation**: Sparsity regularizers Wang et al. (2020); Fang et al. (2023); Liu et al. (2017); Friedman et al. (2010) aim to induce sparsity in network parameters, rendering redundant parameters proximate to zero or outright zero, thereby facilitating pruning based on importance criteria. However, in practical applications, we find that sparsity regularizers do not necessarily improve performance and only show significant effects in specific scenarios. Notably, across all experiments utilizing sparsity regularizers (refer to Table 9-21 in the A.5), only $57.30\%$ showcase positive performance improvements with sparsity regularization. Among these techniques, BNScale delivers the most favorable outcomes, having a $77.78\%$ probability of enhancing performance, followed by GroupLASSO with a $65.38\%$ likelihood of improvement. Conversely, GroupNorm and GrowingReg demonstrate lower effectiveness overall, with improvement probabilities of $42.31\%$ and $45.83\%$, respectively. Nonetheless, these methods excel in particular architectural settings. GrowingReg, for instance, excels in the ViT architecture, manifesting notable performance enhancements across all speedup ratios, while other techniques improve ViT performance less than half of the time. GroupNorm, on the other hand, is better suited for VGG models, exhibiting a $66.67\%$ probability of performance enhancement, a significant improvement compared to its performance in other architectures. A drawback of sparsity regularizers is the necessity for meticulous tuning tailored to each scenario, *i.e.,* the optimal hyperparameters vary across diverse model-dataset configurations. Please refer to the A.2.4 for more details of the hyperparameters.

### Q5: How consistent are the CIFAR rankings with the ImageNet rankings?

**Observation**: In comparison to pruning CIFAR100-trained models, pruning ImageNet-trained models (of the same architecture as CIFAR100-trained ones) typically results in greater accuracy deterioration. Meanwhile, ImageNet-trained models are more sensitive to the speedup ratios. However, for the same model architecture, the leaderboard rankings on CIFAR are highly consistent with those on ImageNet (see Table 11, 14, 19-20 in A.5 for details). For example, when pruning ResNet models, MagnitudeL1, MagnitudeL2, BNScale, and Taylor methods consistently rank within the top five on both CIFAR100 and ImageNet, whereas LAMP and Hrank consistently rank low on the list. These observations indicate that pruning methods showcase a degree of consistency across datasets with the same model. In situations where computational resources are constrained, the utilization of smaller datasets for assessing pruning methodologies, followed by the application of the top-ranked techniques to prune larger models, emerges as a viable approach.

## 4.2 MORE DISCUSSIONS

**Local pruning, global pruning *versus* protected global pruning.** Table 4 presents the results of ResNet50 pruned by MagnitudeL2 on ImageNet, with the three pruning schemes. Results on more models are provided in A.5. From these results, we get the following two conclusions.

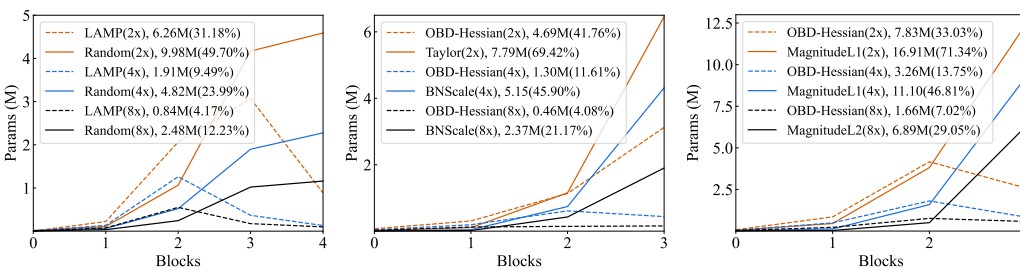

(a) Parameter curves on VGG19.  (b) Parameter curves on ResNet18.  (c) Parameter curves on ResNet50.

Figure 2: The parameter curves of different models pruned by different importance criteria on CIFAR100 dataset. Details can be referred to in Tables 11, 14, and 17 in the A.5.

(1) At low speedup ratios, global pruning and the proposed protected global pruning exhibit comparable performance, both surpassing local pruning. It can be attributed to the assumption in local pruning, which assigns equal importance to each group and applies the same pruning ratio to each group, overlooking group differences. To address this, some previous works such as He & Han (2018); Li et al. (2016); Luo et al. (2017); Yu et al. (2018) propose sensitivity analysis in order to estimate the pruning ratio that should be applied to particular layers (Molchanov et al., 2019).

Table 4: Results of ResNet50 pruned by MagnitudeL2 importance with three pruning schemes.

| Speed Up | Prune Strategy | Base | Pruned | Δ Acc | Parameters |
|---|---|---|---|---|---|
| *2x* | global+protect | 76.128 | 73.684 | -2.444 | 18.26 **M** (*71.44%*) |
|  | global prune | 76.128 | 73.028 | -3.100 | 18.43 **M** (*72.12%*) |
|  | local prune | 76.128 | 70.984 | -5.144 | 12.99 **M** (*50.85%*) |
| *3x* | global+protect | 76.128 | 71.805 | -4.323 | 14.37 **M** (*56.23%*) |
|  | global prune | 76.128 | 63.486 | -12.642 | 14.20 **M** (*55.57%*) |
|  | local prune | 76.128 | 69.168 | -6.96 | 8.77 **M** (*34.31%*) |
| *4x* | global+protect | 76.128 | 69.866 | -6.262 | 11.88 **M** (*46.49%*) |
|  | global prune | 76.128 | 56.068 | -20.06 | 12.38 **M** (*48.46%*) |
|  | local prune | 76.128 | 66.050 | -10.078 | 6.63 **M** (*25.94%*) |

(2) In contrast, at higher speedup ratios, protected global pruning outperforms both local pruning and global pruning. An examination of the network architecture after global pruning uncovers instances of layer collapse, where nearly all channels of a network layer are eliminated, rendering the network untrainable and severely impairing performance.

**Parameters *versus* FLOPS.** Some prior works Park et al. (2020); Lee et al. (2020); Alizadeh et al. (2022); Gonzalez-Carabarin et al. (2022); Rachwan et al. (2022); Salehinejad & Valaee (2021); Dubey et al. (2018); Hu et al. (2016); Tan & Motani (2020) employ the number of parameters as the performance metric of pruned models. Here we discuss the correlation between parameters and the computation cost, *i.e.,* FLOPS. As evidenced in Table 2 and 3, different methods may yield significantly different numbers of pruned parameters at the identical speedup ratios, indicating unequal contributions of various parameters to the computational overhead. Specifically, as depicted in Figure 2, at the same speedup ratio, models with fewer total parameters have more parameters in their initial blocks. Here we provide a brief theoretical insight on this phenomenon. For a convolutional layer $\mathbf{W} \in \mathbb{R}^{\hat{N}NK^2}$, the input tensor $\mathbf{I} \in \mathbb{R}^{NHW}$, and the output tensor $\mathbf{O} \in \mathbb{R}^{\hat{N}\hat{H}\hat{W}}$, the computational complexity of this layer can be denoted by $\mathbf{I}$ is $O(N\hat{N}\hat{H}\hat{W}K^2)$. The average computational contribution of each parameter is thus $O(\hat{H}\hat{W})$, which implies a positive linear correlation between the computation overhead and the feature map resolution. As CNN models usually progressively scale down the spatial resolution of feature maps as layers deepen, the method that prioritizes the reduction of shallow parameters can effectively decrease computational costs while minimizing the parameter count. However, the same method can exhibit varying preferences when pruning different architectures. For example, OBD-Hessian removes numerous parameters when pruning ResNet18 and ResNet50 (see Table 2 and Table 9 to Table 14 in A.5), indicating a preference for pruning later layers. However, when pruning VGG19, it removes much fewer parameters, suggesting a focus on earlier layers (see Table 15 to Table 17 in A.5).

**CNNs *versus* ViTs.** CNNs and ViTs present diverse characteristics and demonstrate distinct behaviours in structural pruning. For instance, in the case of CNN architectures, owing to the aforementioned

Table 5: Results of MagnitudeL2 with different speedup ratios on ImageNet and CIFAR100.

| Dataset | Model | Speed Up | Base | Pruned | △ Acc | Dataset | Model | Speed Up | Base | Pruned | △ Acc |
|---|---|---|---|---|---|---|---|---|---|---|---|
| **ImageNet** | *ViT-Small* (22.05M) | *2x* | 78.588 | 68.316 | -10.272 | **CIFAR100** | *VGG19* (20.09M) | *2x* | 73.87 | 73.22 | -0.65 |
| | | *3x* | 78.588 | 62.342 | -16.246 | | | *4x* | 73.87 | 71.95 | -1.92 |
| | | *4x* | 78.588 | 59.282 | -19.506 | | | *8x* | 73.87 | 64.96 | -8.91 |
| | *ResNet-50* (25.56M) | *2x* | 76.128 | 73.684 | -2.444 | | *ResNet-50* (23.70M) | *2x* | 78.35 | 78.32 | -0.03 |
| | | *3x* | 76.128 | 71.805 | -4.323 | | | *4x* | 78.35 | 77.98 | -0.37 |
| | | *4x* | 76.128 | 69.866 | -6.262 | | | *8x* | 78.35 | 76.38 | -1.97 |
| | *ResNet-18* (11.69M) | *2x* | 69.758 | 67.502 | -2.256 | | *ResNet-18* (11.23M) | *2x* | 75.61 | 75.72 | +0.11 |
| | | *3x* | 69.758 | 63.284 | -6.474 | | | *4x* | 75.61 | 74.01 | -1.60 |
| | | *4x* | 69.758 | 60.438 | -9.32 | | | *8x* | 75.61 | 71.87 | -3.74 |

relationship between parameters and computational overhead, there can be large differences in the number of parameters pruned by different methods at the identical speedup ratios (see Table 2 and other CNN experiments in A.5). However, for the ViT-small experiments in Table 3, the differences in the number of parameters among different methods at the same speedup ratio are small. This phenomenon arises from the fixed-shaped flattened tensors that characterize the output feature maps of ViT, ensuring a consistent contribution of parameters to computational overhead across distinct layers. Therefore, in contrast to CNN, using the number of parameters as the performance metric for pruning ViTs can also lead to reliable conclusions, thanks to the nearly linear correlation between parameters and computational cost.

Another crucial aspect of Vision Transformers (ViTs) is the intricate interconnection of the patch embedding layer with other layers. The output dimension of the patch embedding layer plays a pivotal role in determining the input dimension for all attention layers, making ViT pruning particularly sensitive to this layer. Additionally, ViT necessitates pruning same dimensions for different attention heads, thereby increasing the implementation complexity. Moreover, in comparison to ResNet50, ViT-small with a similar model size, suffers from more accuracy loss. As depicted in Table 5, at the same speedup ratio, the accuracy loss of ViT-small is several times greater than that of ResNet50. The experimental result aligns with the general consensus in prior literature Chen et al. (2021); Rao et al. (2021); Song et al. (2022); Hou & Kung (2022); Kuznedelev et al. (2024) that ViT models are *harder to compress* while preserving accuracy compared to their classic convolutional counterparts. For further comparisons, please consult Table 3 and Table 20 in the A.5.

**Method applicability.** The applicability of different pruning methods exhibits considerable variance. Most methods are tailored for CNNs, presenting obstacles when adapting them to alternative architectural designs. For instance, HRank Lin et al. (2020) determines channel importance based on the rank of feature map corresponding to each channel, which is incompatible with architectures like ViTs. Since ViTs output flattened tensors, pruning through this method is unfeasible. Analogous challenges emerge with Batch Normalization (BN)-based techniques (Liu et al., 2017; You et al., 2019; Zhuang et al., 2020; Ye et al., 2018; Kang & Han, 2020), which rely on batch normalization layers for importance score. Consequently, these methods can not be directly applied to architectures without batch normalization layers. In contrast to the previously mentioned approaches, techniques based on weight normalization (Li et al., 2016; He et al., 2018; Lee et al., 2020) and weight similarity (Wang et al., 2019b; He et al., 2019; Wang et al., 2021b; Yvinec et al., 2022) exhibit minimal constraints and can be seamlessly integrated into diverse architectural frameworks.

## 5 CONCLUSION AND FUTURE WORK

In this work, we present, to the best of our knowledge, the first comprehensive structural pruning benchmark, PruningBench. PruningBench systematically evaluates 16 existing structural pruning methods on a wide array of models and tasks, yielding a handful of interesting findings which are not explored previously. Furthermore, PruningBench is designed as an expandable package that standardizes experimental settings and eases the integration of new algorithmic implementations. In the future work, we will make a broader study on structural pruning evaluation, covering more advanced models like language models, diffusion models, GNNs, *etc*.

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

# A APPENDIX

## A.1 RELATED WORK

PruningBench categorizes current structural pruning methods into importance criteria and sparsity regularizers. Importance criteria assess the importance of filters within a neural network, identifying redundant filters or channels which need to be pruned, whereas sparsity regularizers aim to learn structured sparse networks by imposing sparse constraints on loss functions and zeroing out certain weights during training.

The sparsity regularizers can be applied to Batch Normalization (BN) parameters He & Xiao (2023) if the networks contain batch normalization layers (Liu et al., 2017; You et al., 2019; Zhuang et al., 2020; Ye et al., 2018), and then the BN parameters are used to indicate the pruning decision of structures such as channels or filters. Sparsity regularizers can also be directly applied to filters (He & Xiao, 2023; Wang et al., 2020; Fang et al., 2023; Friedman et al., 2010; Wen et al., 2016). Group Lasso regularization Friedman et al. (2010); Wen et al. (2016) is commonly used to sparsify filters in a structured manner. Growing Regularization (GREG) Wang et al. (2020) exploits regularization under a growing penalty and uses two algorithms. More recently, GroupNorm (Fang et al., 2023) promotes sparsity across all grouped layers, convering convolutions, batch normalizations and fully-connected layers.

Importance criteria can be divided into two approaches: ***data-free*** methods and ***data-driven*** methods. Data-free methods rely solely on the existing weight information of the network and do not depend on input data, making their pruning results deterministic. These methods can be classified into four categories: weight-norm, weight-correlation, BN-based, and random. Weight-norm methods Li et al. (2016); Lee et al. (2020); He et al. (2018) prune based on the norms of weight values. Representative works, such as MagnitudeL1 (Li et al., 2016), MagnitudeL2 (Li et al., 2016), and LAMP (Lee et al., 2020), consider filters with smaller norms to have weak activation, thus contributing less to the final classification decision (He & Xiao, 2023). Weight-correlation methods He et al. (2019); Wang et al. (2019b; 2021b); Yvinec et al. (2021; 2022) prune based on the relationships between weight values. For instance, FPGM identifies filters close to the geometric median to be redundant, as they represent common information shared by all filters in the same layer and should be removed. BN-based methods Wang et al. (2020); Fang et al. (2023); Friedman et al. (2010); Wen et al. (2016) prune based on the weights of BN layers. BNScale (Liu et al., 2017) directly uses the scaling parameter $\gamma$ of the BN layer to compute the importance scores, while Kang *et al.* Kang & Han (2020) also consider shifting parameters $\beta$. Random methods (Mittal et al., 2018) perform pruning in a random manner.

In contrast, data-driven methods are pruning techniques that require input samples, making their results non-deterministic and dependent on the quality of the input data. These methods can be categorized into activation-based and gradient-based approaches. Activation-based methods (He

Table 6: The performance under different pruning steps on the CIFAR100 dataset, including accuracy change, parameter count and FLOPs. The experiments aim to yield a fourfold speedup (*i.e.,* maintaining 25% of the original FLOPs) for ResNet18.

| Method | Steps | Base | Pruned | $\triangle$ Acc | Parameters | FLOPs |
|---|---|---|---|---|---|---|
| BNScale | 10 | 75.60 | 72.90 | -2.7 | 3.08 **M** (*27.48%*) | 93.57 **M** (*16.81%*) |
| | 50 | 75.60 | 74.00 | -2.06 | 5.03 **M** (*44.84%*) | 135.84 **M** (*24.40%*) |
| | 400 | 75.60 | 73.68 | -1.92 | 5.15 **M** (*45.90%*) | 137.45 **M** (*24.69%*) |
| MagnitudeL2 | 10 | 75.60 | 73.36 | -2.24 | 3.19 **M** (*28.42%*) | 102.98 **M** (*18.50%*) |
| | 50 | 75.60 | 74.44 | -1.66 | 4.34 **M** (*38.72%*) | 138.05 **M** (*24.80%*) |
| | 400 | 75.60 | 74.01 | -1.59 | 4.43 **M** (*39.51%*) | 138.65 **M** (*24.91%*) |

et al., 2017; Lin et al., 2020; Luo et al., 2017; Dubey et al., 2018; Sui et al., 2021; Hu et al., 2016; Tan & Motani, 2020) utilize activation maps for pruning. For example, CP He et al. (2017) and HRank Lin et al. (2020) evaluate channel importance of current layer using reconstruction error and activation map decomposition, respectively. ThiNet (Luo et al., 2017), on the other hand, uses activation maps from the next layer to guide the pruning of the current layer. Gradient-based methods (Wang et al., 2019a; Fang et al., 2023; Molchanov et al., 2019; LeCun et al., 1989; Hassibi & Stork, 1992) rely on gradients or Hessian information to perform pruning. Methods that rely solely on gradients, such as Taylor-FO Molchanov et al. (2019) and Mol-16 (Molchanov et al., 2016), can obtain importance scores from backpropagation without requiring additional memory. Conversely, hessian-based methods, such as OBD-Hessian (Fang et al., 2023) and OBD-C (Wang et al., 2019a), require calculating second-order derivatives, which are computationally prohibitive.

## A.2 HYPERPARAMETERS

### A.2.1 PRUNING STEP

A larger pruning step value allows for finer control over FLOPs. Table 6 demonstrates that setting the pruning step to 10 often results in excessive pruning, leading to FLOPs significantly below the target and a corresponding decrease in accuracy. While the accuracy changes are similar for pruning steps set to 50 and 400, the latter offers more precise FLOPs control. Therefore, we chose 400 pruning steps for the leaderboard experiments.

### A.2.2 HYPERPARAMETERS OF PRETRAINING

As mentioned in the main text, the models for the CIFAR100 experiments are pretrained by us, while the experiments on other datasets utilize publicly available pretrained weights. For the CIFAR100 CNN experiments, we pretrain the models (ResNet18, ResNet50, VGG19) for 200 epochs using SGD with an initial learning rate of 0.1. The learning rate decreases by a factor of 10 at the 120th, 150th, and 180th epochs. We set the batch size to 128 and the weight decay to $5 \times 10^{-4}$.

### A.2.3 HYPERPARAMETERS OF FINETUNING STAGE

*CNN experiments.* For the CNN experiments on CIFAR100, we use SGD with an initial learning rate of 0.01. The learning rate is reduced to one-tenth of its original value every 20 epochs after 60 epochs, until fine-tuning concludes at the 100th epochs. We set the batch size to 128, the weight decay to $5 \times 10^{-4}$, and the Nesterov momentum to 0.9. For the CNN experiments on ImageNet, we adjust the learning rate to 0.1 and the weight decay to $1 \times 10^{-4}$. The learning rate is reduced by a factor of 10 at the 30th and 60th epochs, with fine-tuning concluding at the 90th epochs.

*ViT-small experiments.* For ViT-small experiments on ImageNet, we adopt AdamW as the optimizer. The batch size is set to 128 and the weight decay to 0.3. Various data augmentation techniques, as mentioned by Touvron et al. (Touvron et al., 2021), are employed. Due to the slow convergence and sensitivity to the learning rate of ViT-small, we use different learning rates for different speedup ratios. Specifically, for a speedup ratio of 2, the learning rate is set to $1.5 \times 10^{-5}$, while for speedup ratios of 3 and 4, the learning rate is set to $1.5 \times 10^{-4}$. The cosine annealing schedule is used for learning rate decay and the fine-tuning finishes at the 90th epochs.

Table 7: The optimal hyperparameters for the sparsity regularizers across different tasks. $\lambda$ denotes the regularization coefficient, $\eta$ is the learning rate for sparse learning, and $\delta$ is the delta coefficient in GrowingReg (Wang et al., 2020). $*$ indicates that the $\eta$ value for the ViT-small model is identical to the learning rate used during its finetuning stage and is adjusted based on the speedup ratio.

| Method | Task | | $\lambda$ | $\eta$ | $\delta$ |
| | Model | Dataset | | | |
| --- | --- | --- | --- | --- | --- |
| *GroupLASSO* (for MagnitudeL2) | VGG19 | CIFAR100 | 0.00001 | 0.001 | – |
| | ResNet18 | CIFAR100 | 0.0005 | 0.005 | – |
| | ResNet50 | CIFAR100 | 0.0001 | 0.005 | – |
| | ResNet18 | ImageNet | 0.00005 | 0.005 | – |
| | ResNet50 | ImageNet | 0.0005 | 0.01 | – |
| | ViT-small | ImageNet | 0.0001 | $*$ | – |
| | YOLOv8 | COCO | 0.0001 | 0.001 | – |
| *GroupLASSO* (for BNScale) | VGG19 | CIFAR100 | 0.0005 | 0.005 | – |
| | ResNet18 | CIFAR100 | 0.00005 | 0.01 | – |
| | ResNet50 | CIFAR100 | 0.00005 | 0.01 | – |
| | ResNet18 | ImageNet | 0.0005 | 0.005 | – |
| | ResNet50 | ImageNet | 0.0005 | 0.01 | – |
| | ViT-small | ImageNet | 0.0001 | $*$ | – |
| | YOLOv8 | COCO | 0.0005 | 0.001 | – |
| *GroupNorm* (for MagnitudeL2) | VGG19 | CIFAR100 | 0.00001 | 0.005 | – |
| | ResNet18 | CIFAR100 | 0.0001 | 0.005 | – |
| | ResNet50 | CIFAR100 | 0.0001 | 0.005 | – |
| | ResNet18 | ImageNet | 0.00005 | 0.01 | – |
| | ResNet50 | ImageNet | 0.0005 | 0.005 | – |
| | ViT-small | ImageNet | 0.0005 | $*$ | – |
| | YOLOv8 | COCO | 0.0001 | 0.01 | – |
| *BNScale* (for BNScale) | VGG19 | CIFAR100 | 0.0005 | 0.005 | – |
| | ResNet18 | CIFAR100 | 0.0001 | 0.01 | – |
| | ResNet50 | CIFAR100 | 0.00001 | 0.01 | – |
| | ResNet18 | ImageNet | 0.0001 | 0.01 | – |
| | ResNet50 | ImageNet | 0.00005 | 0.01 | – |
| | YOLOv8 | COCO | 0.00001 | 0.005 | – |
| *GrowingReg* (for MagnitudeL2) | VGG19 | CIFAR100 | 0.0001 | 0.001 | 0.00001 |
| | ResNet18 | CIFAR100 | 0.0005 | 0.01 | 0.0001 |
| | ResNet50 | CIFAR100 | 0.0001 | 0.001 | 0.00001 |
| | ResNet18 | ImageNet | 0.0001 | 0.005 | 0.00005 |
| | ResNet50 | ImageNet | 0.00005 | 0.01 | 0.00001 |
| | ViT-small | ImageNet | 0.0005 | $*$ | 0.0001 |
| | YOLOv8 | COCO | 0.0001 | 0.005 | 0.00005 |

*YOLOv8 experiments.* For YOLOv8 experiments on COCO, we utilize SGD with a learning rate of 0.01. The learning rate scheduler initiates a warmup phase followed by linear decay until the completion of fine-tuning at the 100th epoch. We configure the batch size to 128, the weight decay to $5 \times 10^{-4}$, and the Nesterov momentum to 0.937.

### A.2.4 HYPERPARAMETERS OF SPARSIFYING STAGE.

Sparsity regularizers require case-by-case tuning of their hyperparameters for optimal sparse learning. Table 7 presents the optimal hyperparameters across different tasks. Other hyperparameters, such as epochs, batch size and weight decay, are consistent with those used in the finetuning stage.

### A.3 UNIFIED INTERFACES

### A.3.1 INTERFACE FOR IMPORTANCE CRITERIA.

PruningBench categorizes the network layers, where each kind of layer requires a different pruning scheme, corresponding to different importance criteria interfaces that users should implement. Because of the interdependencies among these layers, pruning parameters in one layer necessitates the simultaneous pruning of parameters in other layers that depend on it. Thus, users only need to implement a part of the interfaces based on their algorithm, and PruningBench will extend pruning to the entire group. PruningBench classifies the network layers into the following types:

**convolutional input and output layers.** For a convolutional layer $\mathbf{W} \in \mathbb{R}^{\hat{N}NK^2}$ and $\mathbf{b} \in \mathbb{R}^{\hat{N}}$ (where $\mathbf{W}$ represents the weights and $\mathbf{b}$ is the bias), the input tensor $\mathbf{I} \in \mathbb{R}^{NHW}$, and the output tensor $\mathbf{O} \in \mathbb{R}^{\hat{N}\hat{H}\hat{W}}$. In a convolutional **output** layer, the output channels (filters) are pruned, with the

pruning scheme represented by $\mathbf{W}[k,:,:,:]$ and $\mathbf{b}[k]$. In this context, the importance criteria interface that should be implemented by users is $I(\mathbf{W}) \in \mathbb{R}^{\hat{N}}$, where each element of $I(\mathbf{W})$ signifies the importance score of parameters along the first dimension of $\mathbf{W}$. PruningBench selects indices for pruning based on $I(\mathbf{W})$ and removes them accordingly. Subsequently, PruningBench prunes $\mathbf{b}[k]$ and parameters in other layers that are coupled with it. In contrast, in a convolutional **input** layer, the input channels are pruned, with the pruning scheme denoted as $\mathbf{W}[:,k,:,:]$ (the bias remains unaffected), which implies the second dimension of $\mathbf{W}$ should be pruned.

**linear input and output layers.** A linear layer can be parameterized as $\{\mathbf{W} \in \mathbb{R}^{\hat{N}N}, \mathbf{b} \in \mathbb{R}^{\hat{N}}\}$. Same to convolutional input and output layers. linear layers have distinct pruning schemes for their inputs and outputs, *i.e.,* $\mathbf{W}[k,:]$ and $\mathbf{b}[k]$ for output layers and $\mathbf{W}[:,k]$ for input layers.

**normalization layers.** A normalization layer can be parameterized as $\{\gamma \in \mathbb{R}^{\hat{N}}, \beta \in \mathbb{R}^{\hat{N}}\}$, $\gamma$ and $\beta$ indicate the scale and shift parameters, respectively. Unlike convolutional and linear layers, the inputs and outputs of a normalization layer share the same pruning scheme, *i.e.,* $\gamma[k]$ and $\beta[k]$.

The aforementioned network layers already constitute the majority of modern neural networks. In addition to these, PruningBench also offers interfaces for other network layers such as LSTM layer, multi-head attention layer, embedding layer, *etc.,* providing support for a wide range of architectures and tasks.

By traversing the layers within a group $g$, PruningBench computes importance scores for the layers mentioned above. Note that not all layers need to participate in the importance score calculation, and this can be freely adjusted based on the pruning algorithm. Without loss of generality, we present an example upon CNNs: For implementing filter-wise pruning methods (Li et al., 2016; Rachwan et al., 2022; He et al., 2018; Lin et al., 2018), we only need to consider the pruning schema of the convolutional output layer, $\mathbf{W}[k,:,:,:]$, and compute the importance score $I(\mathbf{W})$. This importance score also represents the importance score of the entire group, *i.e.,* $I(g) = I(\mathbf{W})$, as other layers within the group are not considered. In contrast, channel-wise pruning methods He et al. (2017); Hu et al. (2016); Sui et al. (2021); Hou et al. (2022) calculate importance score for the convolutional input layer. The pruning schema is $(\mathbf{W}[:,k,:,:])$. Batch Normalization (BN) based methods Liu et al. (2017); You et al. (2019); Zhuang et al. (2020); Ye et al. (2018) directly uses the scaling parameter $\gamma$ of the BN layer to compute the importance scores, *i.e.,* $I(g) = I(\gamma)$, while Kang *et al.* Kang & Han (2020) also consider shifting parameters $\beta$.

The aforementioned methods determine the importance of the entire group based on a single layer within the group, whereas other methods consider multiple layers. For instance, some methods He et al. (2019); Wen et al. (2016); Gao et al. (2018); Yvinec et al. (2022) consider both input and output layers. Fang *et al.* (Fang et al., 2023) consider parameters from all layers, including the bias parameters. These methods necessitate computing the importance scores for different layers, all having the same dimensionality. PruningBench will then derive the importance score of the entire group $I(g)$ through dimensionality reduction and normalization.

### A.3.2 INTERFACE FOR SPARSITY REGULARIZER.

The main effort of sparsity regularizer is to design the effective target loss function $\mathcal{L}$ with an advanced penalty term to learn structured sparse networks. In the implementation, PruningBench does not actually add an extra penalty term. Instead, following parameter updates via backpropagation of the loss, PruningBench provides an interface for adjusting the gradients according to the regularization coefficient and parameter weights. This approach exhibits greater versatility. For example, the training objective of the BNScale method Liu et al. (2017) is $\mathcal{L} = \sum_{(x,y)} l(f(x,W),y) + \lambda \sum_{\gamma \in \Gamma} p(\gamma)$, where $(x,y)$ denote the train input and target, $W$ denotes the trainable weights, and $\gamma$ denotes the scaling factor for each batch normalization layer. The first sum-term corresponds to the normal training loss. $p(\cdot)$ is a sparsity-induced penalty on the scaling factors, and $\lambda$ is the regularization coefficient. If we choose $p(\gamma) = |\gamma|$, then this regularization term can be modified to operate on the gradients, *i.e.,* $\nabla W = \nabla W + \lambda * |\gamma|$. By directly manipulating the gradients, other sparsity regularizers can also be easily implemented through the PruningBench interface.

Table 8: Leaderboard of ResNet50 on CIFAR100 at three different speedup ratios. Global pruning strategy is adapted.

| Speed Up | Method | | Rank | Base | Pruned | △ Acc | Pruning Ratio | Step Time | Reg Time |
|---|---|---|---|---|---|---|---|---|---|
| | Importance | Regularizer | | | | | | | |
| 2x | OBD-C* | N/A | 1 | 78.35 | 78.67 | +0.32 | 16.64 **M** (*70.19%*) | 7.471s | N/A |
| | MagnitudeL1 | N/A | 2 | 78.35 | 78.36 | +0.01 | 16.98 **M** (*71.62%*) | 0.137s | N/A |
| | FPGM | N/A | 3 | 78.35 | 78.32 | -0.03 | 15.18 **M** (*64.04%*) | 0.163s | N/A |
| | MagnitudeL2 | N/A | 4 | 78.35 | 78.20 | -0.15 | 16.62 **M** (*70.10%*) | 0.136s | N/A |
| | ThiNet* | N/A | 4 | 78.35 | 78.20 | -0.15 | 16.64 **M** (*70.19%*) | 33.516s | N/A |
| | BNScale | N/A | 5 | 78.35 | 78.07 | -0.28 | 15.96 **M** (*67.32%*) | 0.140s | N/A |
| | Taylor* | N/A | 6 | 78.35 | 77.92 | -0.43 | 16.62 **M** (*70.11%*) | 3.725s | N/A |
| | Random* | N/A | 7 | 78.35 | 77.72 | -0.63 | 11.82 **M** (*49.88%*) | 0.104s | N/A |
| | CP* | N/A | 8 | 78.35 | 77.53 | -0.82 | 7.09 **M** (*29.93%*) | 2m51s | N/A |
| | HRank* | N/A | 9 | 78.35 | 77.31 | -1.04 | 7.53 **M** (*31.76%*) | 34m30s | N/A |
| | OBD-Hessian* | N/A | 10 | 78.35 | 77.07 | -1.28 | 7.64 **M** (*32.21%*) | 5m5s | N/A |
| | LAMP | N/A | 11 | 78.35 | 75.44 | -2.91 | 16.23 **M** (*68.46%*) | 0.151s | N/A |
| | MagnitudeL2 | GroupLASSO | 1 | 78.35 | 78.49 | +0.14 | 16.20 **M** (*68.35%*) | 0.136s | 3m5s |
| | MagnitudeL2 | GrowingReg | 2 | 78.35 | 78.38 | +0.03 | 16.21 **M** (*68.39%*) | 0.136s | 3m1s |
| | BNScale | BNScale | 3 | 78.35 | 78.22 | -0.13 | 16.85 **M** (*71.10%*) | 0.140s | 2m14s |
| | MagnitudeL2 | GroupNorm | 4 | 78.35 | 78.05 | -0.30 | 15.10 **M** (*63.71%*) | 0.136s | 3m7s |
| | BNScale | GroupLASSO | 5 | 78.35 | 77.97 | -0.38 | 16.25 **M** (*68.55%*) | 0.140s | 2m38s |
| 4x | BNScale | N/A | 1 | 78.35 | 78.11 | -0.24 | 10.50 **M** (*44.31%*) | 0.140s | N/A |
| | MagnitudeL1 | N/A | 2 | 78.35 | 78.02 | -0.33 | 11.12 **M** (*46.91%*) | 0.137s | N/A |
| | MagnitudeL2 | N/A | 3 | 78.35 | 77.67 | -0.68 | 10.76 **M** (*45.39%*) | 0.136s | N/A |
| | FPGM | N/A | 4 | 78.35 | 77.63 | -0.72 | 9.98 **M** (*42.11%*) | 0.163s | N/A |
| | Taylor* | N/A | 5 | 78.35 | 77.50 | -0.85 | 5.46 **M** (*23.02%*) | 3.725s | N/A |
| | ThiNet* | N/A | 6 | 78.35 | 77.44 | -0.91 | 5.23 **M** (*22.07%*) | 33.516s | N/A |
| | OBD-C* | N/A | 7 | 78.35 | 77.32 | -1.03 | 5.80 **M** (*24.48%*) | 7.471s | N/A |
| | Random* | N/A | 8 | 78.35 | 77.05 | -1.30 | 6.12 **M** (*25.81%*) | 0.104s | N/A |
| | CP* | N/A | 9 | 78.35 | 75.78 | -2.57 | 2.54 **M** (*10.71%*) | 2m51s | N/A |
| | OBD-Hessian* | N/A | 10 | 78.35 | 75.33 | -3.02 | 3.38 **M** (*14.26%*) | 5m5s | N/A |
| | LAMP | N/A | 11 | 78.35 | 74.32 | -4.03 | 6.24 **M** (*26.33%*) | 0.151s | N/A |
| | HRank* | N/A | 12 | 78.35 | 72.06 | -6.29 | 1.63 **M** (*6.87%*) | 34m30s | N/A |
| | BNScale | BNScale | 1 | 78.35 | 77.79 | -0.56 | 10.74 **M** (*45.30%*) | 0.140s | 2m14s |
| | MagnitudeL2 | GrowingReg | 2 | 78.35 | 77.73 | -0.62 | 10.65 **M** (*44.93%*) | 0.136s | 3m1s |
| | BNScale | GroupLASSO | 3 | 78.35 | 77.71 | -0.64 | 10.73 **M** (*45.25%*) | 0.140s | 2m38s |
| | MagnitudeL2 | GroupLASSO | 4 | 78.35 | 77.69 | -0.66 | 10.67 **M** (*45.02%*) | 0.136s | 3m5s |
| | MagnitudeL2 | GroupNorm | 5 | 78.35 | 77.48 | -0.87 | 9.72 **M** (*40.99%*) | 0.136s | 3m7s |
| 8x | MagnitudeL1 | N/A | 1 | 78.35 | 76.48 | -1.87 | 7.00 **M** (*29.52%*) | 0.137s | N/A |
| | BNScale | N/A | 2 | 78.35 | 76.31 | -2.04 | 6.76 **M** (*28.53%*) | 0.140s | N/A |
| | Random* | N/A | 3 | 78.35 | 76.12 | -2.23 | 3.17 **M** (*13.36%*) | 0.104s | N/A |
| | FPGM | N/A | 4 | 78.35 | 76.08 | -2.27 | 6.68 **M** (*28.18%*) | 0.163s | N/A |
| | MagnitudeL2 | N/A | 5 | 78.35 | 76.06 | -2.29 | 7.06 **M** (*29.78%*) | 0.136s | N/A |
| | OBD-C* | N/A | 6 | 78.35 | 75.44 | -2.91 | 2.43 **M** (*10.25%*) | 7.471s | N/A |
| | Taylor* | N/A | 7 | 78.35 | 75.41 | -2.94 | 1.89 **M** (*7.99%*) | 3.725s | N/A |
| | ThiNet* | N/A | 8 | 78.35 | 74.93 | -3.42 | 1.54 **M** (*6.48%*) | 33.516s | N/A |
| | OBD-Hessian* | N/A | 9 | 78.35 | 73.65 | -4.70 | 1.33 **M** (*5.60%*) | 5m5s | N/A |
| | LAMP | N/A | 10 | 78.35 | 73.01 | -5.34 | 3.58 **M** (*15.08%*) | 0.151s | N/A |
| | CP* | N/A | 11 | 78.35 | 72.61 | -5.74 | 0.98 **M** (*4.15%*) | 2m51s | N/A |
| | HRank* | N/A | 12 | 78.35 | 16.61 | -61.74 | 0.40 **M** (*1.67%*) | 34m30s | N/A |
| | MagnitudeL2 | GroupLASSO | 1 | 78.35 | 76.87 | -1.48 | 6.66 **M** (*28.09%*) | 0.136s | 3m5s |
| | MagnitudeL2 | GrowingReg | 2 | 78.35 | 76.68 | -1.67 | 6.69 **M** (*28.22%*) | 0.136s | 3m1s |
| | BNScale | BNScale | 3 | 78.35 | 76.41 | -1.94 | 6.56 **M** (*27.67%*) | 0.140s | 2m14s |
| | MagnitudeL2 | GroupNorm | 4 | 78.35 | 75.81 | -2.54 | 6.64 **M** (*28.00%*) | 0.136s | 3m7s |
| | BNScale | GroupLASSO | 5 | 78.35 | 75.67 | -2.68 | 6.55 **M** (*27.65%*) | 0.140s | 2m38s |

## A.4 PUBLIC LEADERBOARDS

PruningBench currently maintains 13 leaderboards: 9 for CNN classification tasks on CIFAR, covering three different models each evaluated with three pruning strategies; 3 for ImageNet tasks, featuring two ResNet models and ViT-small; and 1 for the YOLOv8 network on the COCO task. These leaderboards are detailed in Tables 8 to Tables 20. Based on these data, we can derive many conclusions and patterns. In addition to the conclusions discussed in the main text, other findings can be observed. For instance, in the YOLO experiments, the performance differences among various pruning methods are minimal. In contrast, other architectures exhibit significant differences, suggesting that the YOLO architecture is more stable for pruning. PruningBench provides various filtering and calculation features and is continually benchmarking more models, facilitating researchers in discovering more valuable findings.

## A.5 SUPPLEMENTARY RESULTS

Table 9: Leaderboard of ResNet50 on CIFAR100 at three different speedup ratios. Local pruning strategy is adapted.

| Speed Up | Method | | Rank | Base | Pruned | Δ Acc | Pruning Ratio | Step Time | Reg Time |
| --- | --- | --- | --- | --- | --- | --- | --- | --- | --- |
| | Importance | Regularizer | | | | | | | |
| 2x | ThiNet* | N/A | 1 | 78.35 | 78.27 | -0.08 | 11.90 M (50.19%) | 36.354s | N/A |
| | FPGM | N/A | 2 | 78.35 | 78.21 | -0.14 | 11.90 M (50.19%) | 0.187s | N/A |
| | MagnitudeL2 | N/A | 3 | 78.35 | 78.17 | -0.18 | 11.90 M (50.19%) | 0.239s | N/A |
| | CP* | N/A | 3 | 78.35 | 78.17 | -0.18 | 11.90 M (50.19%) | 2m47s | N/A |
| | LAMP | N/A | 4 | 78.35 | 78.16 | -0.19 | 11.90 M (50.19%) | 0.165s | N/A |
| | HRank* | N/A | 5 | 78.35 | 78.12 | -0.23 | 11.90 M (50.19%) | 34m25s | N/A |
| | OBD-C* | N/A | 6 | 78.35 | 78.10 | -0.25 | 11.90 M (50.19%) | 7.622s | N/A |
| | MagnitudeL1 | N/A | 7 | 78.35 | 78.08 | -0.27 | 11.90 M (50.19%) | 0.160s | N/A |
| | BNScale | N/A | 7 | 78.35 | 78.08 | -0.27 | 11.90 M (50.19%) | 0.162s | N/A |
| | Taylor* | N/A | 8 | 78.35 | 77.85 | -0.50 | 11.90 M (50.19%) | 3.755s | N/A |
| | OBD-Hessian* | N/A | 9 | 78.35 | 77.78 | -0.57 | 11.90 M (50.19%) | 5m5s | N/A |
| | Random* | N/A | 10 | 78.35 | 77.64 | -0.71 | 11.90 M (50.19%) | 0.105s | N/A |
| | BNScale | BNScale | 1 | 78.35 | 78.20 | -0.15 | 11.90 M (50.19%) | 0.162s | 2m14s |
| | MagnitudeL2 | GrowingReg | 2 | 78.35 | 78.13 | -0.22 | 11.90 M (50.19%) | 0.239s | 3m |
| | MagnitudeL2 | GroupLASSO | 3 | 78.35 | 78.10 | -0.25 | 11.90 M (50.19%) | 0.239s | 3m4s |
| | BNScale | GroupLASSO | 4 | 78.35 | 77.81 | -0.54 | 11.90 M (50.19%) | 0.162s | 2m38s |
| | MagnitudeL2 | GroupNorm | 5 | 78.35 | 77.61 | -0.74 | 11.90 M (50.19%) | 0.239s | 3m7s |
| 4x | MagnitudeL1 | N/A | 1 | 78.35 | 78.02 | -0.33 | 5.90 M (24.89%) | 0.160s | N/A |
| | MagnitudeL2 | N/A | 2 | 78.35 | 77.71 | -0.64 | 5.90 M (24.89%) | 0.239s | N/A |
| | OBD-C* | N/A | 3 | 78.35 | 77.49 | -0.86 | 5.90 M (24.89%) | 7.622s | N/A |
| | HRank* | N/A | 4 | 78.35 | 77.44 | -0.91 | 5.90 M (24.89%) | 34m25s | N/A |
| | CP* | N/A | 5 | 78.35 | 77.43 | -0.92 | 5.90 M (24.89%) | 2m47s | N/A |
| | BNScale | N/A | 6 | 78.35 | 77.34 | -1.01 | 5.90 M (24.89%) | 0.162s | N/A |
| | LAMP | N/A | 7 | 78.35 | 77.27 | -1.08 | 5.90 M (24.89%) | 0.165s | N/A |
| | OBD-Hessian* | N/A | 8 | 78.35 | 77.27 | -1.08 | 5.90 M (24.89%) | 5m5s | N/A |
| | FPGM | N/A | 9 | 78.35 | 77.26 | -1.09 | 5.90 M (24.89%) | 0.187s | N/A |
| | ThiNet* | N/A | 10 | 78.35 | 77.09 | -1.26 | 5.90 M (24.89%) | 36.354s | N/A |
| | Taylor* | N/A | 11 | 78.35 | 76.87 | -1.48 | 5.90 M (24.89%) | 3.755s | N/A |
| | Random* | N/A | 12 | 78.35 | 76.30 | -2.05 | 5.90 M (24.89%) | 0.105s | N/A |
| | BNScale | BNScale | 1 | 78.35 | 77.80 | -0.55 | 5.90 M (24.89%) | 0.162s | 2m14s |
| | MagnitudeL2 | GroupLASSO | 2 | 78.35 | 77.51 | -0.84 | 5.90 M (24.89%) | 0.239s | 3m4s |
| | BNScale | GroupLASSO | 3 | 78.35 | 77.84 | -0.51 | 5.90 M (24.89%) | 0.162s | 2m38s |
| | MagnitudeL2 | GroupNorm | 4 | 78.35 | 77.27 | -1.08 | 5.90 M (24.89%) | 0.239s | 3m7s |
| | MagnitudeL2 | GrowingReg | 5 | 78.35 | 77.32 | -1.03 | 5.90 M (24.89%) | 0.239s | 3m |
| 8x | BNScale | N/A | 1 | 78.35 | 76.96 | -1.39 | 2.99 M (12.61%) | 0.162s | N/A |
| | MagnitudeL1 | N/A | 2 | 78.35 | 76.88 | -1.47 | 2.99 M (12.61%) | 0.160s | N/A |
| | MagnitudeL2 | N/A | 3 | 78.35 | 76.80 | -1.55 | 2.99 M (12.61%) | 0.239s | N/A |
| | FPGM | N/A | 4 | 78.35 | 76.73 | -1.62 | 2.99 M (12.61%) | 0.187s | N/A |
| | OBD-Hessian* | N/A | 5 | 78.35 | 76.49 | -1.86 | 2.99 M (12.61%) | 5m5s | N/A |
| | LAMP | N/A | 6 | 78.35 | 76.34 | -2.01 | 2.99 M (12.61%) | 0.165s | N/A |
| | CP* | N/A | 7 | 78.35 | 76.21 | -2.14 | 2.99 M (12.61%) | 2m47s | N/A |
| | Taylor* | N/A | 8 | 78.35 | 76.12 | -2.23 | 2.99 M (12.61%) | 3.755s | N/A |
| | OBD-C* | N/A | 9 | 78.35 | 76.05 | -2.30 | 2.99 M (12.61%) | 7.622s | N/A |
| | ThiNet* | N/A | 10 | 78.35 | 75.88 | -2.47 | 2.99 M (12.61%) | 36.354s | N/A |
| | Random* | N/A | 11 | 78.35 | 75.86 | -2.49 | 2.99 M (12.61%) | 0.105s | N/A |
| | HRank* | N/A | 12 | 78.35 | 75.31 | -3.04 | 2.99 M (12.61%) | 34m25s | N/A |
| | MagnitudeL2 | GrowingReg | 1 | 78.35 | 76.94 | -1.41 | 2.99 M (12.61%) | 0.239s | 3m |
| | BNScale | BNScale | 2 | 78.35 | 76.85 | -1.50 | 2.99 M (12.61%) | 0.162s | 2m14s |
| | MagnitudeL2 | GroupNorm | 3 | 78.35 | 76.58 | -1.77 | 2.99 M (12.61%) | 0.239s | 3m7s |
| | BNScale | GroupLASSO | 4 | 78.35 | 76.39 | -1.96 | 2.99 M (12.61%) | 0.162s | 2m38s |
| | MagnitudeL2 | GroupLASSO | 5 | 78.35 | 76.35 | -2.00 | 2.99 M (12.61%) | 0.239s | 3m4s |

Table 10: Leaderboard of ResNet50 on CIFAR100 at three different speedup ratios. Global pruning with 10% group-wise protection is adapted.

| Speed Up | Method | | Rank | Base | Pruned | Δ Acc | Pruning Ratio | Step Time | Reg Time |
|---|---|---|---|---|---|---|---|---|---|
| | Importance | Regularizer | | | | | | | |
| 2x | OBD-C* | N/A | 1 | 78.35 | 78.68 | +0.33 | 16.45 M (69.39%) | 7.559s | N/A |
| | Taylor* | N/A | 2 | 78.35 | 78.51 | +0.16 | 16.65 M (70.24%) | 3.740s | N/A |
| | FPGM | N/A | 3 | 78.35 | 78.37 | +0.02 | 15.37 M (64.84%) | 0.163s | N/A |
| | MagnitudeL2 | N/A | 4 | 78.35 | 78.32 | -0.03 | 16.63 M (70.17%) | 0.136s | N/A |
| | BNScale | N/A | 5 | 78.35 | 78.30 | -0.05 | 15.96 M (67.32%) | 0.141s | N/A |
| | ThiNet* | N/A | 6 | 78.35 | 78.14 | -0.21 | 15.19 M (64.06%) | 33.619s | N/A |
| | Random* | N/A | 7 | 78.35 | 77.97 | -0.38 | 11.78 M (49.70%) | 0.104s | N/A |
| | CP* | N/A | 8 | 78.35 | 77.80 | -0.55 | 7.15 M (30.15%) | 2m51s | N/A |
| | MagnitudeL1 | N/A | 9 | 78.35 | 77.62 | -0.73 | 16.91 M (71.34%) | 0.137s | N/A |
| | OBD-Hessian* | N/A | 10 | 78.35 | 77.26 | -1.09 | 7.83 M (33.03%) | 5m5s | N/A |
| | LAMP | N/A | 11 | 78.35 | 76.26 | -2.09 | 16.21 M (68.37%) | 0.150s | N/A |
| | HRank* | N/A | 12 | 78.35 | 76.13 | -2.22 | 6.47 M (27.29%) | 34m32s | N/A |
| | MagnitudeL2 | GroupLASSO | 1 | 78.35 | 78.73 | +0.38 | 16.51 M (69.66%) | 0.136s | 3m5s |
| | BNScale | BNScale | 2 | 78.35 | 78.36 | +0.01 | 15.97 M (67.37%) | 0.141s | 2m14s |
| | MagnitudeL2 | GroupNorm | 3 | 78.35 | 78.30 | -0.05 | 15.03 M (63.41%) | 0.136s | 3m7s |
| | BNScale | GroupLASSO | 4 | 78.35 | 78.24 | -0.11 | 15.86 M (66.90%) | 0.141s | 2m38s |
| | MagnitudeL2 | GrowingReg | 5 | 78.35 | 77.99 | -0.36 | 16.61 M (70.06%) | 0.136s | 3m1s |
| 4x | FPGM | N/A | 1 | 78.35 | 78.02 | -0.33 | 10.23 M (43.16%) | 0.163s | N/A |
| | MagnitudeL2 | N/A | 2 | 78.35 | 77.98 | -0.37 | 10.71 M (45.19%) | 0.136s | N/A |
| | BNScale | N/A | 3 | 78.35 | 77.90 | -0.45 | 10.53 M (44.41%) | 0.141s | N/A |
| | MagnitudeL1 | N/A | 4 | 78.35 | 77.82 | -0.53 | 11.10 M (46.81%) | 0.137s | N/A |
| | Taylor* | N/A | 5 | 78.35 | 77.69 | -0.66 | 5.47 M (23.09%) | 3.740s | N/A |
| | OBD-C* | N/A | 6 | 78.35 | 77.51 | -0.84 | 5.84 M (24.64%) | 7.559s | N/A |
| | Random* | N/A | 7 | 78.35 | 77.41 | -0.94 | 5.95 M (25.11%) | 0.104s | N/A |
| | ThiNet* | N/A | 8 | 78.35 | 77.23 | -1.12 | 4.72 M (19.91%) | 33.619s | N/A |
| | CP* | N/A | 9 | 78.35 | 75.68 | -2.67 | 2.65 M (11.18%) | 2m51s | N/A |
| | LAMP | N/A | 10 | 78.35 | 75.52 | -2.83 | 5.93 M (25.03%) | 0.150s | N/A |
| | OBD-Hessian* | N/A | 11 | 78.35 | 75.49 | -2.86 | 3.26 M (13.75%) | 5m5s | N/A |
| | HRank* | N/A | 12 | 78.35 | 73.76 | -4.59 | 1.69 M (7.11%) | 34m32s | N/A |
| | BNScale | BNScale | 1 | 78.35 | 78.16 | -0.19 | 10.37 M (43.75%) | 0.141s | 2m14s |
| | MagnitudeL2 | GroupLASSO | 2 | 78.35 | 78.01 | -0.34 | 10.79 M (45.53%) | 0.136s | 3m5s |
| | BNScale | GroupLASSO | 3 | 78.35 | 77.90 | -0.45 | 10.76 M (45.38%) | 0.141s | 2m38s |
| | MagnitudeL2 | GroupNorm | 4 | 78.35 | 77.88 | -0.47 | 9.84 M (41.51%) | 0.136s | 3m7s |
| | MagnitudeL2 | GrowingReg | 5 | 78.35 | 77.86 | -0.49 | 10.77 M (45.43%) | 0.136s | 3m1s |
| 8x | MagnitudeL1 | N/A | 1 | 78.35 | 76.99 | -1.36 | 6.82 M (28.77%) | 0.137s | N/A |
| | MagnitudeL2 | N/A | 2 | 78.35 | 76.38 | -1.97 | 6.89 M (29.05%) | 0.136s | N/A |
| | Random* | N/A | 3 | 78.35 | 76.13 | -2.22 | 2.98 M (12.57%) | 0.104s | N/A |
| | FPGM | N/A | 4 | 78.35 | 75.93 | -2.42 | 7.16 M (30.20%) | 0.163 | N/A |
| | BNScale | N/A | 5 | 78.35 | 75.81 | -2.54 | 6.69 M (28.22%) | 0.141s | N/A |
| | OBD-C* | N/A | 6 | 78.35 | 75.78 | -2.57 | 2.35 M (9.92%) | 7.559s | N/A |
| | Taylor* | N/A | 7 | 78.35 | 75.38 | -2.97 | 1.98 M (8.34%) | 3.740s | N/A |
| | ThiNet* | N/A | 8 | 78.35 | 75.29 | -3.06 | 1.58 M (6.68%) | 33.619s | N/A |
| | OBD-Hessian* | N/A | 9 | 78.35 | 74.49 | -3.86 | 1.66 M (7.02%) | 5m5s | N/A |
| | LAMP | N/A | 10 | 78.35 | 73.48 | -4.87 | 3.62 M (15.27%) | 0.150s | N/A |
| | CP* | N/A | 11 | 78.35 | 72.39 | -5.96 | 0.97 M (4.07%) | 2m51s | N/A |
| | HRank* | N/A | 12 | 78.35 | 70.54 | -7.81 | 0.64 M (2.69%) | 34m32s | N/A |
| | MagnitudeL2 | GrowingReg | 1 | 78.35 | 76.39 | -1.96 | 7.00 M (29.52%) | 0.136s | 3m1s |
| | MagnitudeL2 | GroupLASSO | 2 | 78.35 | 76.27 | -2.08 | 7.09 M (29.90%) | 0.136s | 3m5s |
| | MagnitudeL2 | GroupNorm | 3 | 78.35 | 75.93 | -2.42 | 7.18 M (30.28%) | 0.136s | 3m7s |
| | BNScale | GroupLASSO | 4 | 78.35 | 75.60 | -2.75 | 7.19 M (30.32%) | 0.141s | 2m38s |
| | BNScale | BNScale | 5 | 78.35 | 75.47 | -2.88 | 6.90 M (29.12%) | 0.141s | 2m14s |

Table 11: Leaderboard of ResNet18 on CIFAR100 at three different speedup ratios. Global pruning strategy is adapted.

| Speed Up | Method | | Rank | Base | Pruned | △ Acc | Pruning Ratio | Step Time | Reg Time |
|---|---|---|---|---|---|---|---|---|---|
| | **Importance** | **Regularizer** | | | | | | | |
| 2x | FPGM | N/A | 1 | 75.61 | 75.89 | +0.28 | 8.51 **M** (75.83%) | 0.051s | N/A |
| | OBD-C* | N/A | 2 | 75.61 | 75.88 | +0.27 | 7.83 **M** (69.79%) | 5.212s | N/A |
| | Taylor* | N/A | 3 | 75.61 | 75.73 | +0.12 | 7.77 **M** (69.23%) | 2.005s | N/A |
| | MagnitudeL2 | N/A | 4 | 75.61 | 75.72 | +0.11 | 7.55 **M** (67.25%) | 0.375s | N/A |
| | BNScale | N/A | 5 | 75.61 | 75.60 | -0.01 | 7.72 **M** (68.81%) | 0.120s | N/A |
| | ThiNet* | N/A | 6 | 75.61 | 75.49 | -0.12 | 7.53 **M** (67.10%) | 5.705s | N/A |
| | HRank* | N/A | 7 | 75.61 | 75.47 | -0.14 | 4.40 **M** (39.21%) | 8m55s | N/A |
| | CP* | N/A | 8 | 75.61 | 75.38 | -0.23 | 7.39 **M** (65.88%) | 44.892s | N/A |
| | MagnitudeL1 | N/A | 9 | 75.61 | 75.22 | -0.39 | 7.47 **M** (66.62%) | 0.124s | N/A |
| | OBD-Hessian* | N/A | 10 | 75.61 | 74.83 | -0.78 | 5.01 **M** (44.67%) | 1m48s | N/A |
| | Random* | N/A | 11 | 75.61 | 74.15 | -1.46 | 5.68 **M** (50.64%) | 0.048s | N/A |
| | LAMP | N/A | 12 | 75.61 | 73.64 | -1.97 | 6.84 **M** (60.98%) | 0.056s | N/A |
| | MagnitudeL2 | GroupLASSO | 1 | 75.61 | 76.05 | +0.44 | 7.55 **M** (67.30%) | 0.375s | 1m29s |
| | BNScale | GroupLASSO | 2 | 75.61 | 76.05 | +0.44 | 7.77 **M** (69.29%) | 0.120s | 47.132s |
| | BNScale | BNScale | 3 | 75.61 | 76.01 | +0.40 | 7.70 **M** (68.64%) | 0.120s | 36.281s |
| | MagnitudeL2 | GrowingReg | 4 | 75.61 | 75.76 | +0.15 | 7.88 **M** (70.23%) | 0.375s | 1m31s |
| | MagnitudeL2 | GroupNorm | 5 | 75.61 | 75.56 | -0.05 | 7.76 **M** (69.19%) | 0.375s | 1m26s |
| 4x | MagnitudeL2 | N/A | 1 | 75.61 | 74.54 | -1.07 | 4.45 **M** (39.70%) | 0.375s | N/A |
| | ThiNet* | N/A | 2 | 75.61 | 73.98 | -1.63 | 2.87 **M** (25.56%) | 5.705s | N/A |
| | BNScale | N/A | 3 | 75.61 | 73.88 | -1.73 | 5.15 **M** (45.90%) | 0.120s | N/A |
| | MagnitudeL1 | N/A | 4 | 75.61 | 73.83 | -1.78 | 4.68 **M** (41.74%) | 0.124s | N/A |
| | Taylor* | N/A | 5 | 75.61 | 73.79 | -1.82 | 3.22 **M** (28.72%) | 2.005s | N/A |
| | CP* | N/A | 6 | 75.61 | 73.78 | -1.83 | 3.51 **M** (31.28%) | 44.892s | N/A |
| | OBD-C* | N/A | 7 | 75.61 | 73.77 | -1.84 | 4.17 **M** (37.16%) | 5.212s | N/A |
| | FPGM | N/A | 8 | 75.61 | 73.62 | -1.99 | 5.27 **M** (46.95%) | 0.051s | N/A |
| | Random* | N/A | 9 | 75.61 | 72.33 | -3.28 | 2.99 **M** (26.68%) | 0.048s | N/A |
| | OBD-Hessian* | N/A | 10 | 75.61 | 71.18 | -4.43 | 1.14 **M** (10.14%) | 1m48s | N/A |
| | HRank* | N/A | 11 | 75.61 | 70.66 | -4.95 | 0.95 **M** (8.46%) | 8m55s | N/A |
| | LAMP | N/A | 12 | 75.61 | 66.04 | -9.57 | 3.26 **M** (29.09%) | 0.056s | N/A |
| | MagnitudeL2 | GroupNorm | 1 | 75.61 | 74.37 | -1.24 | 4.07 **M** (36.29%) | 0.375s | 1m26s |
| | MagnitudeL2 | GrowingReg | 2 | 75.61 | 74.16 | -1.45 | 4.44 **M** (39.59%) | 0.375s | 1m31s |
| | MagnitudeL2 | GroupLASSO | 3 | 75.61 | 74.15 | -1.46 | 4.45 **M** (39.67%) | 0.375s | 1m29s |
| | BNScale | GroupLASSO | 4 | 75.61 | 73.99 | -1.62 | 5.12 **M** (45.63%) | 0.120s | 47.132s |
| | BNScale | BNScale | 5 | 75.61 | 73.81 | -1.80 | 4.85 **M** (43.23%) | 0.120s | 36.281s |
| 8x | MagnitudeL2 | N/A | 1 | 75.61 | 71.63 | -3.98 | 2.35 **M** (20.92%) | 0.375s | N/A |
| | OBD-C* | N/A | 2 | 75.61 | 71.15 | -4.46 | 1.28 **M** (11.42%) | 5.212s | N/A |
| | BNScale | N/A | 3 | 75.61 | 71.01 | -4.60 | 2.50 **M** (22.31%) | 0.120s | N/A |
| | MagnitudeL1 | N/A | 4 | 75.61 | 70.96 | -4.65 | 2.12 **M** (18.93%) | 0.124s | N/A |
| | CP* | N/A | 5 | 75.61 | 70.79 | -4.82 | 1.05 **M** (9.39%) | 44.892s | N/A |
| | ThiNet* | N/A | 6 | 75.61 | 70.49 | -5.12 | 0.75 **M** (6.65%) | 5.705s | N/A |
| | Taylor* | N/A | 7 | 75.61 | 70.18 | -5.43 | 0.76 **M** (6.80%) | 2.005s | N/A |
| | Random* | N/A | 8 | 75.61 | 69.80 | -5.81 | 1.31 **M** (11.72%) | 0.048s | N/A |
| | LAMP | N/A | 9 | 75.61 | 69.12 | -6.49 | 0.46 **M** (4.07%) | 0.056s | N/A |
| | OBD-Hessian* | N/A | 10 | 75.61 | 65.57 | -10.04 | 0.37 **M** (3.33%) | 1m48s | N/A |
| | FPGM | N/A | 11 | 75.61 | 59.80 | -15.81 | 2.97 **M** (26.51%) | 0.051s | N/A |
| | HRank* | N/A | 12 | 75.61 | 51.61 | -24.00 | 0.27 **M** (2.37%) | 8m55s | N/A |
| | MagnitudeL2 | GroupNorm | 1 | 75.61 | 72.10 | -3.51 | 2.20 **M** (19.65%) | 0.375s | 1m26s |
| | MagnitudeL2 | GroupLASSO | 2 | 75.61 | 71.66 | -3.95 | 2.38 **M** (21.23%) | 0.375s | 2m29s |
| | MagnitudeL2 | GrowingReg | 3 | 75.61 | 71.57 | -4.04 | 2.34 **M** (20.87%) | 0.375s | 1m31s |
| | BNScale | GroupLASSO | 4 | 75.61 | 71.50 | -4.11 | 2.49 **M** (22.18%) | 0.120s | 47.132s |
| | BNScale | BNScale | 5 | 75.61 | 71.44 | -4.17 | 2.36 **M** (21.00%) | 0.120s | 36.281s |

Table 12: Leaderboard of ResNet18 on CIFAR100 at three different speedup ratios. Local pruning strategy is adapted.

| Speed Up | Method | | Rank | Base | Pruned | Δ Acc | Pruning Ratio | Step Time | Reg Time |
| | Importance | Regularizer | | | | | | | |
|---|---|---|---|---|---|---|---|---|---|
| 2x | ThiNet* | N/A | 1 | 75.61 | 75.30 | -0.31 | 5.64 **M** (*50.26%*) | 10.076s | N/A |
| | MagnitudeL1 | N/A | 2 | 75.61 | 74.91 | -0.70 | 5.64 **M** (*50.26%*) | 0.244s | N/A |
| | HRank* | N/A | 3 | 75.61 | 74.81 | -0.80 | 5.64 **M** (*50.26%*) | 11m | N/A |
| | OBD-C* | N/A | 4 | 75.61 | 74.75 | -0.86 | 5.64 **M** (*50.26%*) | 4.193s | N/A |
| | BNScale | N/A | 5 | 75.61 | 74.70 | -0.91 | 5.64 **M** (*50.26%*) | 0.261s | N/A |
| | FPGM | N/A | 6 | 75.61 | 74.70 | -0.91 | 5.64 **M** (*50.26%*) | 0.365s | N/A |
| | Taylor* | N/A | 7 | 75.61 | 74.67 | -0.94 | 5.64 **M** (*50.26%*) | 1.895s | N/A |
| | CP* | N/A | 8 | 75.61 | 74.57 | -1.04 | 5.64 **M** (*50.26%*) | 46.008s | N/A |
| | OBD-Hessian* | N/A | 9 | 75.61 | 74.56 | -1.05 | 5.64 **M** (*50.26%*) | 1m47s | N/A |
| | MagnitudeL2 | N/A | 10 | 75.61 | 74.40 | -1.21 | 5.64 **M** (*50.26%*) | 0.048s | N/A |
| | LAMP | N/A | 11 | 75.61 | 74.26 | -1.35 | 5.64 **M** (*50.26%*) | 0.092s | N/A |
| | Random* | N/A | 12 | 75.61 | 74.23 | -1.38 | 5.64 **M** (*50.26%*) | 0.222s | N/A |
| | MagnitudeL2 | GroupLASSO | 1 | 75.61 | 75.06 | -0.55 | 5.64 **M** (*50.26%*) | 0.048s | 1m29s |
| | BNScale | BNScale | 2 | 75.61 | 74.94 | -0.67 | 5.64 **M** (*50.26%*) | 0.261s | 51.869s |
| | BNScale | GroupLASSO | 3 | 75.61 | 74.94 | -0.67 | 5.64 **M** (*50.26%*) | 0.261s | 1m20s |
| | MagnitudeL2 | GrowingReg | 4 | 75.61 | 74.70 | -0.91 | 5.64 **M** (*50.26%*) | 0.048s | 1m31s |
| | MagnitudeL2 | GroupNorm | 5 | 75.61 | 74.40 | -1.21 | 5.64 **M** (*50.26%*) | 0.048s | 1m32s |
| 4x | MagnitudeL2 | N/A | 1 | 75.61 | 73.43 | -2.18 | 2.77 **M** (*24.71%*) | 0.048s | N/A |
| | FPGM | N/A | 2 | 75.61 | 73.35 | -2.26 | 2.77 **M** (*24.71%*) | 0.365s | N/A |
| | Taylor* | N/A | 3 | 75.61 | 73.29 | -2.32 | 2.77 **M** (*24.71%*) | 1.895s | N/A |
| | OBD-Hessian* | N/A | 4 | 75.61 | 73.28 | -2.33 | 2.77 **M** (*24.71%*) | 1m47s | N/A |
| | BNScale | N/A | 5 | 75.61 | 73.17 | -2.44 | 2.77 **M** (*24.71%*) | 0.261s | N/A |
| | CP* | N/A | 6 | 75.61 | 73.11 | -2.50 | 2.77 **M** (*24.71%*) | 46.008s | N/A |
| | MagnitudeL1 | N/A | 7 | 75.61 | 73.09 | -2.52 | 2.77 **M** (*24.71%*) | 0.244s | N/A |
| | HRank* | N/A | 8 | 75.61 | 72.91 | -2.70 | 2.77 **M** (*24.71%*) | 11m | N/A |
| | ThiNet* | N/A | 9 | 75.61 | 72.82 | -2.79 | 2.77 **M** (*24.71%*) | 10.076s | N/A |
| | OBD-C* | N/A | 10 | 75.61 | 72.61 | -3.00 | 2.77 **M** (*24.71%*) | 4.193s | N/A |
| | LAMP | N/A | 11 | 75.61 | 72.01 | -3.60 | 2.77 **M** (*24.71%*) | 0.092s | N/A |
| | Random* | N/A | 12 | 75.61 | 71.97 | -3.64 | 2.77 **M** (*24.71%*) | 0.222s | N/A |
| | MagnitudeL2 | GroupNorm | 1 | 75.61 | 73.37 | -2.24 | 2.77 **M** (*24.71%*) | 0.048s | 1m32s |
| | BNScale | BNScale | 2 | 75.61 | 73.24 | -2.37 | 2.77 **M** (*24.71%*) | 0.261s | 51.869s |
| | MagnitudeL2 | GrowingReg | 3 | 75.61 | 73.17 | -2.44 | 2.77 **M** (*24.71%*) | 0.048s | 1m31s |
| | MagnitudeL2 | GroupLASSO | 4 | 75.61 | 73.13 | -2.48 | 2.77 **M** (*24.71%*) | 0.048s | 1m29s |
| | BNScale | GroupLASSO | 5 | 75.61 | 72.93 | -2.68 | 2.77 **M** (*24.71%*) | 0.261s | 1m20s |
| 8x | MagnitudeL2 | N/A | 1 | 75.61 | 72.01 | -3.60 | 1.44 **M** (*12.83%*) | 0.048s | N/A |
| | MagnitudeL1 | N/A | 2 | 75.61 | 71.60 | -4.01 | 1.44 **M** (*12.83%*) | 0.244s | N/A |
| | OBD-Hessian* | N/A | 3 | 75.61 | 71.60 | -4.01 | 1.44 **M** (*12.83%*) | 1m47s | N/A |
| | ThiNet* | N/A | 4 | 75.61 | 71.51 | -4.10 | 1.44 **M** (*12.83%*) | 10.076s | N/A |
| | FPGM | N/A | 5 | 75.61 | 71.13 | -4.48 | 1.44 **M** (*12.83%*) | 0.365s | N/A |
| | BNScale | N/A | 6 | 75.61 | 71.11 | -4.50 | 1.44 **M** (*12.83%*) | 0.261s | N/A |
| | Taylor* | N/A | 7 | 75.61 | 70.91 | -4.70 | 1.44 **M** (*12.83%*) | 1.895s | N/A |
| | CP* | N/A | 8 | 75.61 | 70.85 | -4.76 | 1.44 **M** (*12.83%*) | 46.008s | N/A |
| | OBD-C* | N/A | 9 | 75.61 | 70.78 | -4.83 | 1.44 **M** (*12.83%*) | 4.193s | N/A |
| | HRank* | N/A | 10 | 75.61 | 70.60 | -5.01 | 1.44 **M** (*12.83%*) | 11m | N/A |
| | Random* | N/A | 11 | 75.61 | 69.89 | -5.72 | 1.44 **M** (*12.83%*) | 0.222s | N/A |
| | LAMP | N/A | 12 | 75.61 | 66.84 | -8.77 | 1.44 **M** (*12.83%*) | 0.092s | N/A |
| | MagnitudeL2 | GroupLASSO | 1 | 75.61 | 72.44 | -3.17 | 1.44 **M** (*12.83%*) | 0.048s | 1m29s |
| | MagnitudeL2 | GrowingReg | 2 | 75.61 | 71.94 | -3.67 | 1.44 **M** (*12.83%*) | 0.048s | 1m31s |
| | BNScale | GroupLASSO | 3 | 75.61 | 71.66 | -3.95 | 1.44 **M** (*12.83%*) | 0.261s | 1m20s |
| | MagnitudeL2 | GroupNorm | 4 | 75.61 | 71.60 | -4.01 | 1.44 **M** (*12.83%*) | 0.048s | 1m32s |
| | BNScale | BNScale | 5 | 75.61 | 71.15 | -4.46 | 1.44 **M** (*12.83%*) | 0.261s | 51.869s |

Table 13: Leaderboard of ResNet18 on CIFAR100 at three different speedup ratios. Global pruning with 10% group-wise protection is adapted.

| Speed Up | Method | | Rank | Base | Pruned | Δ Acc | Pruning Ratio | Step Time | Reg Time |
|---|---|---|---|---|---|---|---|---|---|
| | **Importance** | **Regularizer** | | | | | | | |
| 2x | Taylor* | N/A | 1 | 75.61 | 75.93 | +0.32 | 7.79 **M** (69.42%) | 1.598s | N/A |
| | MagnitudeL1 | N/A | 2 | 75.61 | 75.80 | +0.19 | 7.47 **M** (66.62%) | 0.058s | N/A |
| | OBD-Hessian* | N/A | 3 | 75.61 | 75.79 | +0.18 | 4.69 **M** (41.76%) | 1m46s | N/A |
| | MagnitudeL2 | N/A | 4 | 75.61 | 75.72 | +0.11 | 7.55 **M** (67.25%) | 0.261s | N/A |
| | ThiNet* | N/A | 5 | 75.61 | 75.72 | +0.11 | 7.56 **M** (67.38%) | 7.815s | N/A |
| | BNScale | N/A | 6 | 75.61 | 75.51 | -0.10 | 7.72 **M** (68.81%) | 0.263s | N/A |
| | CP* | N/A | 7 | 75.61 | 75.49 | -0.12 | 7.44 **M** (66.33%) | 42.944s | N/A |
| | OBD-C* | N/A | 8 | 75.61 | 75.32 | -0.29 | 7.61 **M** (67.81%) | 4.942s | N/A |
| | FPGM | N/A | 9 | 75.61 | 75.16 | -0.45 | 8.21 **M** (73.20%) | 0.049s | N/A |
| | HRank* | N/A | 10 | 75.61 | 74.90 | -0.69 | 5.12 **M** (45.63%) | 9m8s | N/A |
| | Random* | N/A | 11 | 75.61 | 74.20 | -1.41 | 5.52 **M** (49.22%) | 0.047s | N/A |
| | LAMP | N/A | 12 | 75.61 | 73.95 | -1.66 | 6.84 **M** (60.99%) | 0.059s | N/A |
| | BNScale | GroupLASSO | 1 | 75.61 | 76.05 | +0.44 | 7.77 **M** (69.29%) | 0.263s | 1m19s |
| | BNScale | BNScale | 2 | 75.61 | 76.01 | +0.40 | 7.70 **M** (68.64%) | 0.263s | 49.425s |
| | MagnitudeL2 | GrowingReg | 3 | 75.61 | 75.76 | +0.15 | 7.88 **M** (70.23%) | 0.261s | 1m 35s |
| | MagnitudeL2 | GroupLASSO | 4 | 75.61 | 75.57 | -0.04 | 7.55 **M** (67.30%) | 0.261s | 1m33s |
| | MagnitudeL2 | GroupNorm | 5 | 75.61 | 75.56 | -0.05 | 7.76 **M** (69.19%) | 0.261s | 1m36s |
| 4x | MagnitudeL2 | N/A | 1 | 75.61 | 74.01 | -1.60 | 4.43 **M** (39.51%) | 0.261s | N/A |
| | ThiNet* | N/A | 2 | 75.61 | 73.99 | -1.62 | 2.59 **M** (23.12%) | 7.815s | N/A |
| | OBD-C* | N/A | 3 | 75.61 | 73.94 | -1.67 | 4.23 **M** (37.70%) | 4.942s | N/A |
| | Taylor* | N/A | 4 | 75.61 | 73.83 | -1.78 | 2.98 **M** (26.52%) | 1.598s | N/A |
| | BNScale | N/A | 5 | 75.61 | 73.68 | -1.93 | 5.15 **M** (45.90%) | 0.263s | N/A |
| | MagnitudeL1 | N/A | 6 | 75.61 | 73.53 | -2.08 | 4.62 **M** (41.17%) | 0.058s | N/A |
| | FPGM | N/A | 7 | 75.61 | 73.49 | -2.12 | 5.04 **M** (44.91%) | 0.049s | N/A |
| | CP* | N/A | 8 | 75.61 | 73.18 | -2.43 | 3.13 **M** (27.94%) | 42.944s | N/A |
| | OBD-Hessian* | N/A | 9 | 75.61 | 72.72 | -2.89 | 1.30 **M** (11.61%) | 1m46s | N/A |
| | HRank* | N/A | 10 | 75.61 | 72.17 | -3.44 | 1.17 **M** (10.42%) | 9m8s | N/A |
| | Random* | N/A | 11 | 75.61 | 71.85 | -3.76 | 2.69 **M** (23.94%) | 0.047s | N/A |
| | LAMP | N/A | 12 | 75.61 | 70.81 | -4.80 | 3.39 **M** (30.20%) | 0.059s | N/A |
| | MagnitudeL2 | GroupNorm | 1 | 75.61 | 74.37 | -1.24 | 4.07 **M** (36.29%) | 0.261s | 1m36s |
| | MagnitudeL2 | GrowingReg | 2 | 75.61 | 74.16 | -1.45 | 4.44 **M** (39.59%) | 0.261s | 1m35s |
| | MagnitudeL2 | GroupLASSO | 3 | 75.61 | 74.15 | -1.46 | 4.45 **M** (39.67%) | 0.261s | 1m33s |
| | BNScale | GroupLASSO | 4 | 75.61 | 73.99 | -1.62 | 5.12 **M** (45.63%) | 0.263s | 1m19s |
| | BNScale | BNScale | 5 | 75.61 | 73.81 | -1.80 | 4.85 **M** (43.23%) | 0.263s | 49.425s |
| 8x | MagnitudeL2 | N/A | 1 | 75.61 | 71.87 | -3.74 | 2.32 **M** (20.65%) | 0.261s | N/A |
| | BNScale | N/A | 2 | 75.61 | 71.31 | -4.30 | 2.37 **M** (21.17%) | 0.263s | N/A |
| | OBD-C* | N/A | 3 | 75.61 | 71.27 | -4.34 | 1.43 **M** (12.71%) | 4.942s | N/A |
| | MagnitudeL1 | N/A | 4 | 75.61 | 70.51 | -5.10 | 2.27 **M** (20.20%) | 0.058s | N/A |
| | Taylor* | N/A | 5 | 75.61 | 70.34 | -5.27 | 0.78 **M** (6.92%) | 1.598s | N/A |
| | CP* | N/A | 6 | 75.61 | 70.23 | -5.38 | 1.05 **M** (9.34%) | 42.944s | N/A |
| | FPGM | N/A | 7 | 75.61 | 69.87 | -5.74 | 2.82 **M** (25.17%) | 0.049s | N/A |
| | LAMP | N/A | 8 | 75.61 | 69.68 | -5.93 | 0.46 **M** (4.07%) | 0.059s | N/A |
| | Random* | N/A | 9 | 75.61 | 69.48 | -6.13 | 1.34 **M** (11.93%) | 0.047s | N/A |
| | ThiNet* | N/A | 10 | 75.61 | 69.03 | -6.58 | 0.52 **M** (4.64%) | 7.815s | N/A |
| | OBD-Hessian* | N/A | 11 | 75.61 | 68.55 | -7.06 | 0.46 **M** (4.08%) | 1m46s | N/A |
| | HRank* | N/A | 12 | 75.61 | 68.53 | -7.08 | 0.50 **M** (4.48%) | 9m8s | N/A |
| | MagnitudeL2 | GroupNorm | 1 | 75.61 | 72.10 | -3.51 | 2.20 **M** (19.65%) | 0.261s | 1m36s |
| | MagnitudeL2 | GroupLASSO | 2 | 75.61 | 71.66 | -3.95 | 2.38 **M** (21.23%) | 0.261 | 1m33s |
| | MagnitudeL2 | GrowingReg | 3 | 75.61 | 71.57 | -4.04 | 2.34 **M** (20.87%) | 0.261 | 1m35s |
| | BNScale | GroupLASSO | 4 | 75.61 | 71.50 | -4.11 | 2.49 **M** (22.18%) | 0.263 | 1m19s |
| | BNScale | BNScale | 5 | 75.61 | 71.44 | -4.17 | 2.36 **M** (21.00%) | 0.263 | 49.425s |

Table 14: Leaderboard of VGG19 on CIFAR100 at three different speedup ratios. Global pruning strategy is adapted.

| Speed Up | Method | | Rank | Base | Pruned | Δ Acc | Pruning Ratio | Step Time | Reg Time |
| | Importance | Regularizer | | | | | | | |
|---|---|---|---|---|---|---|---|---|---|
| 2x | MagnitudeL2 | N/A | 1 | 73.87 | 73.88 | +0.01 | 7.15 **M** (*35.61%*) | 0.061s | N/A |
| | CP* | N/A | 2 | 73.87 | 73.75 | -0.12 | 5.02 **M** (*25.00%*) | 1m2s | N/A |
| | OBD-C* | N/A | 3 | 73.87 | 73.69 | -0.18 | 7.27 **M** (*36.18%*) | 4.847s | N/A |
| | HRank* | N/A | 4 | 73.87 | 73.68 | -0.19 | 6.27 **M** (*31.20%*) | 11m47s | N/A |
| | MagnitudeL1 | N/A | 5 | 73.87 | 73.65 | -0.22 | 7.25 **M** (*36.10%*) | 0.133s | N/A |
| | LAMP | N/A | 6 | 73.87 | 73.53 | -0.34 | 5.58 **M** (*27.79%*) | 0.070s | N/A |
| | BNScale | N/A | 7 | 73.87 | 73.51 | -0.36 | 7.18 **M** (*35.72%*) | 0.051s | N/A |
| | Taylor* | N/A | 8 | 73.87 | 73.40 | -0.47 | 9.22 **M** (*45.91%*) | 1.605s | N/A |
| | ThiNet* | N/A | 9 | 73.87 | 73.19 | -0.68 | 7.27 **M** (*36.17%*) | 13.880s | N/A |
| | FPGM | N/A | 10 | 73.87 | 73.12 | -0.75 | 7.05 **M** (*35.09%*) | 0.221s | N/A |
| | Random* | N/A | 11 | 73.87 | 72.22 | -1.65 | 10.31 **M** (*51.32%*) | 0.268s | N/A |
| | OBD-Hessian* | N/A | 12 | 73.87 | 71.68 | -2.19 | 8.49 **M** (*42.27%*) | 1m13s | N/A |
| | MagnitudeL2 | GroupLASSO | 1 | 73.87 | 74.16 | +0.29 | 7.12 **M** (*35.44%*) | 0.061s | 1m32s |
| | MagnitudeL2 | GroupNorm | 2 | 73.87 | 73.96 | +0.09 | 6.35 **M** (*31.64%*) | 0.061s | 1m25s |
| | BNScale | BNScale | 3 | 73.87 | 73.98 | -0.11 | 6.33 **M** (*31.49%*) | 0.051s | 36.332s |
| | BNScale | GroupLASSO | 4 | 73.87 | 73.46 | -0.41 | 6.39 **M** (*31.82%*) | 0.051s | 54.597s |
| | MagnitudeL2 | GrowingReg | 5 | 73.87 | 73.34 | -0.53 | 6.35 **M** (*31.64%*) | 0.061s | 1m20s |
| 4x | OBD-C* | N/A | 1 | 73.87 | 72.42 | -1.45 | 2.23 **M** (*11.12%*) | 4.847s | N/A |
| | FPGM | N/A | 2 | 73.87 | 71.79 | -2.08 | 3.08 **M** (*15.34%*) | 0.221s | N/A |
| | Taylor* | N/A | 3 | 73.87 | 71.29 | -2.58 | 3.81 **M** (*18.97%*) | 1.605s | N/A |
| | Random* | N/A | 4 | 73.87 | 71.26 | -2.61 | 4.89 **M** (*24.35%*) | 0.268s | N/A |
| | HRank* | N/A | 5 | 73.87 | 71.19 | -2.68 | 1.51 **M** (*7.52%*) | 11m47s | N/A |
| | ThiNet* | N/A | 6 | 73.87 | 70.77 | -3.10 | 3.11 **M** (*15.46%*) | 13.880s | N/A |
| | CP* | N/A | 7 | 73.87 | 70.37 | -3.50 | 1.81 **M** (*8.99%*) | 1m2s | N/A |
| | LAMP | N/A | 8 | 73.87 | 70.32 | -3.55 | 1.97 **M** (*9.82%*) | 0.070s | N/A |
| | MagnitudeL2 | N/A | 9 | 73.87 | 69.89 | -3.98 | 2.64 **M** (*13.14%*) | 0.061s | N/A |
| | MagnitudeL1 | N/A | 10 | 73.87 | 69.76 | -4.11 | 2.56 **M** (*12.74%*) | 0.133s | N/A |
| | BNScale | N/A | 11 | 73.87 | 69.75 | -4.12 | 3.01 **M** (*14.98%*) | 0.051s | N/A |
| | OBD-Hessian* | N/A | 12 | 73.87 | 68.65 | -5.22 | 3.50 **M** (*17.44%*) | 1m13s | N/A |
| | MagnitudeL2 | GroupNorm | 1 | 73.87 | 72.06 | -1.81 | 3.65 **M** (*18.16%*) | 0.061s | 1m25s |
| | BNScale | GroupLASSO | 2 | 73.87 | 71.96 | -1.91 | 2.95 **M** (*14.69%*) | 0.051s | 54.597s |
| | BNScale | BNScale | 3 | 73.87 | 71.96 | -1.91 | 2.96 **M** (*14.76%*) | 0.051s | 36.332s |
| | MagnitudeL2 | GrowingReg | 4 | 73.87 | 70.35 | -3.52 | 2.67 **M** (*13.30%*) | 0.061s | 1m20s |
| | MagnitudeL2 | GroupLASSO | 5 | 73.87 | 69.41 | -4.46 | 2.65 **M** (*13.17%*) | 0.061s | 1m32s |
| 8x | LAMP | N/A | 1 | 73.87 | 69.91 | -3.96 | 0.84 **M** (*4.17%*) | 0.070s | N/A |
| | OBD-C* | N/A | 2 | 73.87 | 67.73 | -6.14 | 0.80 **M** (*4.00%*) | 4.847s | N/A |
| | Taylor* | N/A | 3 | 73.87 | 67.05 | -6.82 | 2.04 **M** (*10.16%*) | 1.605s | N/A |
| | Random* | N/A | 4 | 73.87 | 65.96 | -7.91 | 2.50 **M** (*12.47%*) | 0.268s | N/A |
| | CP* | N/A | 5 | 73.87 | 65.83 | -8.04 | 0.86 **M** (*4.27%*) | 1m2s | N/A |
| | ThiNet* | N/A | 6 | 73.87 | 65.71 | -8.16 | 1.59 **M** (*7.94%*) | 13.880s | N/A |
| | FPGM | N/A | 7 | 73.87 | 64.24 | -9.63 | 1.77 **M** (*8.83%*) | 0.221s | N/A |
| | OBD-Hessian* | N/A | 8 | 73.87 | 62.10 | -11.77 | 1.81 **M** (*9.00%*) | 1m13s | N/A |
| | MagnitudeL1 | N/A | 9 | 73.87 | 61.20 | -12.67 | 1.40 **M** (*6.99%*) | 0.133s | N/A |
| | MagnitudeL2 | N/A | 10 | 73.87 | 60.59 | -13.28 | 1.44 **M** (*7.15%*) | 0.061s | N/A |
| | BNScale | N/A | 11 | 73.87 | 48.37 | -25.50 | 1.54 **M** (*7.68%*) | 0.051s | N/A |
| | HRank* | N/A | 12 | 73.87 | 0.04 | -73.83 | 0.43 **M** (*2.16%*) | 11m47s | N/A |
| | MagnitudeL2 | GroupLASSO | 1 | 73.87 | 63.26 | -10.61 | 1.44 **M** (*7.14%*) | 0.061s | 1m32s |
| | MagnitudeL2 | GrowingReg | 2 | 73.87 | 62.64 | -11.23 | 1.44 **M** (*7.14%*) | 0.061s | 1m20s |
| | MagnitudeL2 | GroupNorm | 3 | 73.87 | 57.82 | -16.05 | 2.08 **M** (*10.37%*) | 0.061s | 1m25s |
| | BNScale | BNScale | 4 | 73.87 | 48.14 | -25.73 | 1.54 **M** (*7.68%*) | 0.051s | 36.332s |
| | BNScale | GroupLASSO | 5 | 73.87 | 0.01 | -73.86 | 1.52 **M** (*7.58%*) | 0.051s | 54.597s |

Table 15: Leaderboard of VGG19 on CIFAR100 at three different speedup ratios. Local pruning strategy is adapted.

| Speed Up | Method | | Rank | Base | Pruned | Δ Acc | Pruning Ratio | Step Time | Reg Time |
|---|---|---|---|---|---|---|---|---|---|
| | Importance | Regularizer | | | | | | | |
| 2x | MagnitudeL2 | N/A | 1 | 73.87 | 73.13 | -0.74 | 9.95 **M** (*49.51%*) | 0.053s | N/A |
| | BNScale | N/A | 2 | 73.87 | 72.96 | -0.91 | 9.95 **M** (*49.51%*) | 0.279s | N/A |
| | HRank* | N/A | 3 | 73.87 | 72.84 | -1.03 | 9.95 **M** (*49.51%*) | 11m59s | N/A |
| | OBD-C* | N/A | 4 | 73.87 | 72.80 | -1.07 | 9.95 **M** (*49.51%*) | 4.932s | N/A |
| | FPGM | N/A | 5 | 73.87 | 72.72 | -1.15 | 9.95 **M** (*49.51%*) | 0.234s | N/A |
| | LAMP | N/A | 6 | 73.87 | 72.70 | -1.17 | 9.95 **M** (*49.51%*) | 0.335s | N/A |
| | MagnitudeL1 | N/A | 7 | 73.87 | 72.60 | -1.27 | 9.95 **M** (*49.51%*) | 0.054s | N/A |
| | Taylor* | N/A | 8 | 73.87 | 72.50 | -1.37 | 9.95 **M** (*49.51%*) | 1.461s | N/A |
| | OBD-Hessian* | N/A | 9 | 73.87 | 72.45 | -1.42 | 9.95 **M** (*49.51%*) | 1m10s | N/A |
| | ThiNet* | N/A | 10 | 73.87 | 72.38 | -1.49 | 9.95 **M** (*49.51%*) | 16.068s | N/A |
| | CP* | N/A | 11 | 73.87 | 72.36 | -1.51 | 9.95 **M** (*49.51%*) | 54.655s | N/A |
| | Random* | N/A | 12 | 73.87 | 72.19 | -1.68 | 9.95 **M** (*49.51%*) | 0.037s | N/A |
| | MagnitudeL2 | GroupNorm | 1 | 73.87 | 73.14 | -0.73 | 9.95 **M** (*49.51%*) | 0.053s | 1m9s |
| | MagnitudeL2 | GrowingReg | 2 | 73.87 | 73.03 | -0.84 | 9.95 **M** (*49.51%*) | 0.053s | 1m8s |
| | MagnitudeL2 | GroupLASSO | 3 | 73.87 | 72.98 | -0.89 | 9.95 **M** (*49.51%*) | 0.053s | 1m21s |
| | BNScale | BNScale | 4 | 73.87 | 72.82 | -1.05 | 9.95 **M** (*49.51%*) | 0.279s | 34.595s |
| | BNScale | GroupLASSO | 5 | 73.87 | 72.51 | -1.36 | 9.95 **M** (*49.51%*) | 0.279s | 45.508s |
| 4x | Taylor* | N/A | 1 | 73.87 | 71.01 | -2.86 | 4.96 **M** (*24.69%*) | 1.461s | N/A |
| | BNScale | N/A | 2 | 73.87 | 71.01 | -2.86 | 4.96 **M** (*24.69%*) | 0.279s | N/A |
| | FPGM | N/A | 3 | 73.87 | 70.96 | -2.91 | 4.96 **M** (*24.69%*) | 0.234s | N/A |
| | MagnitudeL1 | N/A | 4 | 73.87 | 70.90 | -2.97 | 4.96 **M** (*24.69%*) | 0.054s | N/A |
| | HRank* | N/A | 5 | 73.87 | 70.89 | -2.98 | 4.96 **M** (*24.69%*) | 11m59s | N/A |
| | MagnitudeL2 | N/A | 6 | 73.87 | 70.70 | -3.17 | 4.96 **M** (*24.69%*) | 0.053s | N/A |
| | LAMP | N/A | 7 | 73.87 | 70.70 | -3.17 | 4.96 **M** (*24.69%*) | 0.335s | N/A |
| | Random* | N/A | 8 | 73.87 | 70.31 | -3.56 | 4.96 **M** (*24.69%*) | 0.037s | N/A |
| | OBD-Hessian* | N/A | 9 | 73.87 | 70.30 | -3.57 | 4.96 **M** (*24.69%*) | 1m10s | N/A |
| | OBD-C* | N/A | 10 | 73.87 | 70.11 | -3.76 | 4.96 **M** (*24.69%*) | 4.932s | N/A |
| | CP* | N/A | 11 | 73.87 | 69.93 | -3.94 | 4.96 **M** (*24.69%*) | 54.655s | N/A |
| | ThiNet* | N/A | 12 | 73.87 | 69.78 | -4.09 | 4.96 **M** (*24.69%*) | 16.068s | N/A |
| | MagnitudeL2 | GroupNorm | 1 | 73.87 | 72.06 | -1.81 | 4.96 **M** (*24.69%*) | 0.053s | 1m9s |
| | BNScale | GroupLASSO | 2 | 73.87 | 71.96 | -1.91 | 4.96 **M** (*24.69%*) | 0.279s | 45.508s |
| | BNScale | BNScale | 3 | 73.87 | 71.96 | -1.91 | 4.96 **M** (*24.69%*) | 0.279s | 34.595s |
| | MagnitudeL2 | GrowingReg | 4 | 73.87 | 70.35 | -3.52 | 4.96 **M** (*24.69%*) | 0.053s | 1m8s |
| | MagnitudeL2 | GroupLASSO | 5 | 73.87 | 69.41 | -4.46 | 4.96 **M** (*24.69%*) | 0.053s | 1m21s |
| 8x | MagnitudeL2 | N/A | 1 | 73.87 | 68.19 | -5.68 | 2.50 **M** (*12.44%*) | 0.053s | N/A |
| | MagnitudeL1 | N/A | 2 | 73.87 | 67.67 | -6.20 | 2.50 **M** (*12.44%*) | 0.054s | N/A |
| | FPGM | N/A | 3 | 73.87 | 67.59 | -6.28 | 2.50 **M** (*12.44%*) | 0.234s | N/A |
| | OBD-Hessian* | N/A | 4 | 73.87 | 67.44 | -6.43 | 2.50 **M** (*12.44%*) | 1m10s | N/A |
| | LAMP | N/A | 5 | 73.87 | 67.20 | -6.67 | 2.50 **M** (*12.44%*) | 0.335s | N/A |
| | Taylor* | N/A | 6 | 73.87 | 67.20 | -6.67 | 2.50 **M** (*12.44%*) | 1.461s | N/A |
| | HRank* | N/A | 7 | 73.87 | 67.19 | -6.68 | 2.50 **M** (*12.44%*) | 11m59s | N/A |
| | OBD-C* | N/A | 8 | 73.87 | 66.24 | -7.63 | 2.50 **M** (*12.44%*) | 4.932s | N/A |
| | BNScale | N/A | 9 | 73.87 | 65.95 | -7.92 | 2.50 **M** (*12.44%*) | 0.279s | N/A |
| | Random* | N/A | 10 | 73.87 | 65.40 | -8.47 | 2.50 **M** (*12.44%*) | 0.037s | N/A |
| | CP* | N/A | 11 | 73.87 | 65.05 | -8.82 | 2.50 **M** (*12.44%*) | 54.655s | N/A |
| | ThiNet* | N/A | 12 | 73.87 | 64.99 | -8.88 | 2.50 **M** (*12.44%*) | 16.068s | N/A |
| | MagnitudeL2 | GroupLASSO | 1 | 73.87 | 67.77 | -6.10 | 2.50 **M** (*12.44%*) | 0.053s | 1m21s |
| | MagnitudeL2 | GrowingReg | 2 | 73.87 | 67.59 | -6.28 | 2.50 **M** (*12.44%*) | 0.053s | 1m8s |
| | BNScale | GroupLASSO | 3 | 73.87 | 66.94 | -6.93 | 2.50 **M** (*12.44%*) | 0.279s | 45.508s |
| | BNScale | BNScale | 4 | 73.87 | 66.30 | -7.57 | 2.50 **M** (*12.44%*) | 0.279s | 34.595s |
| | MagnitudeL2 | GroupNorm | 5 | 73.87 | 64.41 | -9.46 | 2.50 **M** (*12.44%*) | 0.053s | 1m9s |

Table 16: Leaderboard of VGG19 on CIFAR100 at three different speedup ratios. Global pruning with 10% group-wise protection is adapted.

| Speed Up | Method | | Rank | Base | Pruned | Δ Acc | Pruning Ratio | Step Time | Reg Time |
| | Importance | Regularizer | | | | | | | |
|---|---|---|---|---|---|---|---|---|---|
| 2x | CP* | N/A | 1 | 73.87 | 74.16 | +0.29 | 4.93 **M** (24.54%) | 1m2s | N/A |
| | HRank* | N/A | 2 | 73.87 | 73.63 | -0.24 | 6.24 **M** (31.08%) | 13m59s | N/A |
| | MagnitudeL1 | N/A | 3 | 73.87 | 73.62 | -0.25 | 7.22 **M** (35.95%) | 0.156s | N/A |
| | FPGM | N/A | 4 | 73.87 | 73.42 | -0.45 | 7.05 **M** (35.09%) | 0.346s | N/A |
| | LAMP | N/A | 5 | 73.87 | 73.32 | -0.55 | 6.26 **M** (31.18%) | 0.063s | N/A |
| | ThiNet* | N/A | 6 | 73.87 | 73.32 | -0.55 | 8.86 **M** (44.11%) | 13.528s | N/A |
| | OBD-C* | N/A | 7 | 73.87 | 73.25 | -0.62 | 7.60 **M** (37.84%) | 5.813s | N/A |
| | MagnitudeL2 | N/A | 8 | 73.87 | 73.22 | -0.65 | 7.14 **M** (35.55%) | 0.199s | N/A |
| | BNScale | N/A | 9 | 73.87 | 73.12 | -0.75 | 7.15 **M** (35.62%) | 0.062s | N/A |
| | Taylor* | N/A | 10 | 73.87 | 73.08 | -0.79 | 9.09 **M** (45.24%) | 1.440s | N/A |
| | Random* | N/A | 11 | 73.87 | 72.75 | -1.12 | 9.98 **M** (49.70%) | 0.172s | N/A |
| | OBD-Hessian* | N/A | 12 | 73.87 | 71.79 | -2.08 | 8.23 **M** (40.96%) | 1m15s | N/A |
| | BNScale | BNScale | 1 | 73.87 | 74.27 | +0.40 | 6.80 **M** (33.84%) | 0.062s | 37.060s |
| | MagnitudeL2 | GroupNorm | 2 | 73.87 | 74.12 | +0.25 | 6.08 **M** (30.26%) | 0.199s | 1m31s |
| | MagnitudeL2 | GrowingReg | 3 | 73.87 | 73.86 | -0.01 | 7.07 **M** (35.18%) | 0.199s | 1m24s |
| | MagnitudeL2 | GroupLASSO | 4 | 73.87 | 73.42 | -0.45 | 7.09 **M** (35.29%) | 0.199s | 1m28s |
| | BNScale | GroupLASSO | 5 | 73.87 | 73.14 | -0.73 | 7.09 **M** (35.29%) | 0.062s | 58.990s |
| 4x | FPGM | N/A | 1 | 73.87 | 72.38 | -1.49 | 3.11 **M** (15.49%) | 0.346s | N/A |
| | LAMP | N/A | 2 | 73.87 | 72.30 | -1.57 | 1.91 **M** (9.49%) | 0.063s | N/A |
| | MagnitudeL2 | N/A | 3 | 73.87 | 71.95 | -1.92 | 2.74 **M** (13.66%) | 0.199s | N/A |
| | MagnitudeL1 | N/A | 4 | 73.87 | 71.89 | -1.98 | 2.64 **M** (13.12%) | 0.156s | N/A |
| | OBD-C* | N/A | 5 | 73.87 | 71.67 | -2.20 | 2.83 **M** (14.07%) | 5.813s | N/A |
| | HRank* | N/A | 6 | 73.87 | 71.61 | -2.26 | 1.47 **M** (7.34%) | 13m59s | N/A |
| | Taylor* | N/A | 7 | 73.87 | 71.37 | -2.50 | 3.76 **M** (18.73%) | 1.440s | N/A |
| | BNScale | N/A | 8 | 73.87 | 71.33 | -2.54 | 3.04 **M** (15.16%) | 0.062s | N/A |
| | ThiNet* | N/A | 9 | 73.87 | 71.17 | -2.70 | 3.95 **M** (19.65%) | 13.528s | N/A |
| | CP* | N/A | 10 | 73.87 | 70.85 | -3.02 | 1.45 **M** (7.21%) | 1m2s | N/A |
| | Random* | N/A | 11 | 73.87 | 70.51 | -3.36 | 4.82 **M** (23.99%) | 0.172s | N/A |
| | OBD-Hessian* | N/A | 12 | 73.87 | 69.12 | -4.75 | 3.82 **M** (19.04%) | 1m15s | N/A |
| | BNScale | BNScale | 1 | 73.87 | 72.34 | -1.53 | 2.67 **M** (13.30%) | 0.062s | 37.060s |
| | BNScale | GroupLASSO | 2 | 73.87 | 72.25 | -1.62 | 2.64 **M** (13.14%) | 0.062s | 58.990s |
| | MagnitudeL2 | GrowingReg | 3 | 73.87 | 71.84 | -2.03 | 2.58 **M** (12.86%) | 0.199s | 1m24s |
| | MagnitudeL2 | GroupLASSO | 4 | 73.87 | 71.68 | -2.19 | 2.59 **M** (12.90%) | 0.199s | 1m28s |
| | MagnitudeL2 | GroupNorm | 5 | 73.87 | 68.59 | -5.28 | 3.31 **M** (16.49%) | 0.199s | 1m31s |
| 8x | LAMP | N/A | 1 | 73.87 | 69.72 | -4.15 | 0.84 **M** (4.17%) | 0.063s | N/A |
| | MagnitudeL1 | N/A | 2 | 73.87 | 68.82 | -5.05 | 1.49 **M** (7.41%) | 0.156s | N/A |
| | OBD-C* | N/A | 3 | 73.87 | 67.88 | -5.99 | 1.51 **M** (7.50%) | 5.813s | N/A |
| | Taylor* | N/A | 4 | 73.87 | 67.35 | -6.52 | 2.00 **M** (9.96%) | 1.440s | N/A |
| | HRank* | N/A | 5 | 73.87 | 67.01 | -6.86 | 0.63 **M** (3.15%) | 13m59s | N/A |
| | ThiNet* | N/A | 6 | 73.87 | 66.40 | -7.47 | 1.75 **M** (8.70%) | 13.528s | N/A |
| | BNScale | N/A | 7 | 73.87 | 66.08 | -7.79 | 1.54 **M** (7.65%) | 0.062s | N/A |
| | Random* | N/A | 8 | 73.87 | 65.69 | -8.18 | 2.48 **M** (12.37%) | 0.172s | N/A |
| | MagnitudeL2 | N/A | 9 | 73.87 | 64.96 | -8.91 | 1.57 **M** (7.83%) | 0.199s | N/A |
| | FPGM | N/A | 10 | 73.87 | 63.97 | -9.90 | 1.78 **M** (8.87%) | 0.346s | N/A |
| | CP* | N/A | 11 | 73.87 | 63.55 | -10.32 | 0.67 **M** (3.34%) | 1m2s | N/A |
| | OBD-Hessian* | N/A | 12 | 73.87 | 63.53 | -10.34 | 1.94 **M** (9.64%) | 1m15s | N/A |
| | BNScale | BNScale | 1 | 73.87 | 68.57 | -5.30 | 1.33 **M** (6.62%) | 0.062s | 37.060s |
| | BNScale | GroupLASSO | 2 | 73.87 | 68.55 | -5.32 | 1.33 **M** (6.60%) | 0.062s | 58.990s |
| | MagnitudeL2 | GroupNorm | 3 | 73.87 | 67.29 | -6.58 | 2.06 **M** (10.24%) | 0.199s | 1m31s |
| | MagnitudeL2 | GrowingReg | 4 | 73.87 | 63.91 | -9.96 | 1.23 **M** (6.15%) | 0.199s | 1m24s |
| | MagnitudeL2 | GroupLASSO | 5 | 73.87 | 63.44 | -10.43 | 1.23 **M** (6.13%) | 0.199s | 1m28s |

Table 17: Leaderboard of YOLOv8 on COCO at three different speedup ratios. Global pruning with 10% group-wise protection is adapted.

| Speed Up | Method | | Rank | Base | Pruned | Δ Acc | Pruning Ratio | Step Time | Reg Time |
|---|---|---|---|---|---|---|---|---|---|
| | Importance | Regularizer | | | | | | | |
| 2x | LAMP | N/A | 1 | 49.993 | 44.464 | -5.529 | 6.81 **M** (*26.27%*) | 3.216s | N/A |
| | MagnitudeL2 | N/A | 2 | 49.993 | 44.380 | -5.613 | 15.08 **M** (*58.24%*) | 2.606s | N/A |
| | OBD-Hessian* | N/A | 3 | 49.993 | 44.327 | -5.666 | 8.62 **M** (*33.28%*) | 15.442s | N/A |
| | BNScale | N/A | 4 | 49.993 | 44.160 | -5.833 | 12.98 **M** (*50.11%*) | 2.992s | N/A |
| | Taylor* | N/A | 5 | 49.993 | 44.087 | -5.906 | 11.48 **M** (*44.32%*) | 13.485s | N/A |
| | MagnitudeL1 | N/A | 6 | 49.993 | 44.004 | -5.989 | 12.98 **M** (*50.11%*) | 2.884s | N/A |
| | ThiNet* | N/A | 7 | 49.993 | 43.401 | -6.592 | 8.74 **M** (*33.72%*) | 8m43s | N/A |
| | FPGM | N/A | 8 | 49.993 | 43.167 | -6.826 | 11.88 **M** (*45.87%*) | 2.145s | N/A |
| | HRank* | N/A | 9 | 49.993 | 43.014 | -6.979 | 6.04 **M** (*23.32%*) | 13m24s | N/A |
| | Random* | N/A | 10 | 49.993 | 42.804 | -7.189 | 12.21 **M** (*47.15%*) | 0.666s | N/A |
| | CP* | N/A | 11 | 49.993 | 42.639 | -7.354 | 7.72 **M** (*29.80%*) | 1m7s | N/A |
| | BNScale | BNScale | 1 | 49.993 | 44.781 | -5.212 | 12.16 **M** (*46.94%*) | 2.992s | 1h24m44s |
| | MagnitudeL2 | GroupLASSO | 2 | 49.993 | 44.753 | -5.24 | 14.68 **M** (*56.69%*) | 2.606s | 2h5m45s |
| | BNScale | GroupLASSO | 3 | 49.993 | 44.541 | -5.452 | 12.41 **M** (*47.91%*) | 2.992s | 1h41m36s |
| | MagnitudeL2 | GrowingReg | 4 | 49.993 | 44.440 | -5.553 | 14.71 **M** (*56.78%*) | 2.606s | 1h59m27s |
| | MagnitudeL2 | GroupNorm | 5 | 49.993 | 44.290 | -5.703 | 14.70 **M** (*56.75%*) | 2.606s | 2h3m45s |
| 3x | MagnitudeL2 | N/A | 1 | 49.993 | 40.644 | -9.349 | 9.91 **M** (*38.24%*) | 2.606s | N/A |
| | LAMP | N/A | 2 | 49.993 | 40.112 | -9.881 | 4.03 **M** (*15.55%*) | 3.216s | N/A |
| | BNScale | N/A | 3 | 49.993 | 39.416 | -10.577 | 8.63 **M** (*33.30%*) | 2.992s | N/A |
| | ThiNet* | N/A | 4 | 49.993 | 39.319 | -10.674 | 8.23 **M** (*31.77%*) | 8m43s | N/A |
| | MagnitudeL1 | N/A | 5 | 49.993 | 39.257 | -10.736 | 8.63 **M** (*33.30%*) | 2.884s | N/A |
| | Taylor* | N/A | 6 | 49.993 | 39.237 | -10.756 | 6.65 **M** (*25.68%*) | 13.485s | N/A |
| | OBD-Hessian* | N/A | 7 | 49.993 | 39.061 | -10.932 | 5.48 **M** (*21.17%*) | 15.442s | N/A |
| | CP* | N/A | 8 | 49.993 | 38.164 | -11.829 | 5.85 **M** (*22.60%*) | 1m7s | N/A |
| | HRank* | N/A | 9 | 49.993 | 37.941 | -12.052 | 4.02 **M** (*15.50%*) | 13m24s | N/A |
| | Random* | N/A | 10 | 49.993 | 37.868 | -12.125 | 7.83 **M** (*30.23%*) | 0.666s | N/A |
| | FPGM | N/A | 11 | 49.993 | 37.523 | -12.470 | 5.07 **M** (*19.57%*) | 2.145s | N/A |
| | MagnitudeL2 | GroupNorm | 1 | 49.993 | 40.853 | -9.140 | 9.56 **M** (*36.91%*) | 2.606s | 2h3m45s |
| | MagnitudeL2 | GrowingReg | 2 | 49.993 | 40.735 | -9.258 | 9.65 **M** (*37.25%*) | 2.606s | 1h59m27s |
| | MagnitudeL2 | GroupLASSO | 3 | 49.993 | 40.526 | -9.467 | 9.58 **M** (*36.98%*) | 2.606s | 2h5m45s |
| | BNScale | BNScale | 4 | 49.993 | 40.101 | -9.892 | 8.23 **M** (*31.77%*) | 2.992s | 1h24m44s |
| | BNScale | GroupLASSO | 5 | 49.993 | 39.635 | -10.358 | 8.46 **M** (*32.65%*) | 2.992s | 1h41m36s |
| 4x | MagnitudeL2 | N/A | 1 | 49.993 | 36.606 | -13.387 | 5.52 **M** (*21.30%*) | 2.606s | N/A |
| | MagnitudeL1 | N/A | 2 | 49.993 | 36.159 | -13.834 | 6.36 **M** (*24.57%*) | 2.884s | N/A |
| | BNScale | N/A | 2 | 49.993 | 36.159 | -13.834 | 6.36 **M** (*24.57%*) | 2.992s | N/A |
| | LAMP | N/A | 4 | 49.993 | 35.976 | -14.017 | 2.90 **M** (*11.20%*) | 3.216s | N/A |
| | Taylor* | N/A | 5 | 49.993 | 35.749 | -14.244 | 4.60 **M** (*17.76%*) | 13.485s | N/A |
| | ThiNet* | N/A | 6 | 49.993 | 35.718 | -14.275 | 5.60 **M** (*21.61%*) | 8m43s | N/A |
| | CP* | N/A | 7 | 49.993 | 35.687 | -14.306 | 4.86 **M** (*18.76%*) | 1m7s | N/A |
| | OBD-Hessian* | N/A | 8 | 49.993 | 35.681 | -14.312 | 3.95 **M** (*15.23%*) | 15.442s | N/A |
| | HRank* | N/A | 9 | 49.993 | 34.265 | -15.728 | 2.59 **M** (*10.01%*) | 13m24s | N/A |
| | FPGM | N/A | 10 | 49.993 | 32.215 | -17.778 | 3.20 **M** (*12.33%*) | 2.145s | N/A |
| | Random* | N/A | 11 | 49.993 | 32.205 | -17.788 | 5.63 **M** (*21.72%*) | 0.666s | N/A |
| | MagnitudeL2 | GroupNorm | 1 | 49.993 | 36.546 | -13.447 | 6.28 **M** (*25.25%*) | 2.606s | 2h3m45s |
| | MagnitudeL2 | GroupLASSO | 2 | 49.993 | 36.488 | -13.505 | 6.57 **M** (*25.38%*) | 2.606s | 2h5m45s |
| | MagnitudeL2 | GrowingReg | 3 | 49.993 | 36.460 | -13.533 | 6.60 **M** (*25.48%*) | 2.606s | 1h59m27s |
| | BNScale | GroupLASSO | 4 | 49.993 | 36.301 | -13.692 | 5.66 **M** (*21.85%*) | 2.992s | 1h41m36s |
| | BNScale | BNScale | 5 | 49.993 | 36.279 | -13.714 | 5.98 **M** (*23.09%*) | 2.992s | 1h24m44s |

Table 18: Leaderboard of ResNet18 on ImageNet at three different speedup ratios. Global pruning with 10% group-wise protection is adapted.

| Speed Up | Method | | Rank | Base | Pruned | Δ Acc | Pruning Ratio | Step Time | Reg Time |
| | Importance | Regularizer | | | | | | | |
|---|---|---|---|---|---|---|---|---|---|
| 2x | MagnitudeL2 | N/A | 1 | 69.758 | 67.724 | -2.034 | 10.52 **M** (90.01%) | 0.038s | N/A |
| | MagnitudeL1 | N/A | 2 | 69.758 | 67.652 | -2.106 | 10.22 **M** (87.41%) | 0.023s | N/A |
| | FPGM | N/A | 3 | 69.758 | 67.642 | -2.116 | 9.54 **M** (81.59%) | 0.029s | N/A |
| | BNScale | N/A | 4 | 69.758 | 67.542 | -2.216 | 8.31 **M** (71.07%) | 0.026s | N/A |
| | OBD-C* | N/A | 5 | 69.758 | 67.319 | -2.439 | 3.95 **M** (33.79%) | 24.096s | N/A |
| | Taylor* | N/A | 6 | 69.758 | 67.220 | -2.538 | 4.59 **M** (39.26%) | 22.487S | N/A |
| | ThiNet* | N/A | 7 | 69.758 | 67.211 | -2.547 | 2.81 **M** (24.05%) | 15.645s | N/A |
| | CP* | N/A | 8 | 69.758 | 67.139 | -2.619 | 1.89 **M** (16.19%) | 2m21s | N/A |
| | OBD-Hessian* | N/A | 9 | 69.758 | 66.934 | -2.824 | 1.49 **M** (12.76%) | 1m45s | N/A |
| | Random* | N/A | 10 | 69.758 | 64.788 | -4.970 | 5.44 **M** (46.57%) | 0.020s | N/A |
| | HRank* | N/A | 11 | 69.758 | 63.834 | -5.924 | 3.20 **M** (27.40%) | 49m53s | N/A |
| | LAMP | N/A | 12 | 69.758 | 58.308 | -11.45 | 1.56 **M** (13.37%) | 0.030s | N/A |
| | MagnitudeL2 | GroupLASSO | 1 | 69.758 | 67.765 | -1.993 | 10.31 **M** (88.20%) | 0.038s | 3h10m44s |
| | BNScale | BNScale | 2 | 69.758 | 67.734 | -2.024 | 17.56 **M** (68.70%) | 0.026s | 1h54m9s |
| | BNScale | GroupLASSO | 3 | 69.758 | 67.376 | -2.382 | 10.47 **M** (89.57%) | 0.026s | 2h41m15s |
| | MagnitudeL2 | GroupNorm | 4 | 69.758 | 67.210 | -2.548 | 8.68 **M** (74.21%) | 0.038s | 3h4m27s |
| | MagnitudeL2 | GrowingReg | 5 | 69.758 | 67.112 | -2.646 | 10.66 **M** (24.71%) | 0.038s | 3h12m21s |
| 3x | BNScale | N/A | 1 | 69.758 | 63.684 | -6.074 | 6.97 **M** (59.59%) | 0.026s | N/A |
| | FPGM | N/A | 2 | 69.758 | 63.582 | -6.176 | 8.26 **M** (70.62%) | 0.029s | N/A |
| | OBD-C* | N/A | 3 | 69.758 | 63.312 | -6.446 | 1.50 **M** (12.87%) | 24.096s | N/A |
| | ThiNet* | N/A | 4 | 69.758 | 63.297 | -6.461 | 0.99 **M** (8.49%) | 15.645s | N/A |
| | MagnitudeL1 | N/A | 5 | 69.758 | 63.284 | -6.474 | 9.20 **M** (78.66%) | 0.023s | N/A |
| | MagnitudeL2 | N/A | 6 | 69.758 | 62.936 | -6.822 | 9.32 **M** (79.69%) | 0.038s | N/A |
| | CP* | N/A | 7 | 69.758 | 62.902 | -6.856 | 0.58 **M** (4.97%) | 2m21s | N/A |
| | Taylor* | N/A | 8 | 69.758 | 62.877 | -6.881 | 1.85 **M** (15.79%) | 22.487S | N/A |
| | OBD-Hessian* | N/A | 9 | 69.758 | 61.022 | -8.736 | 0.73 **M** (6.26%) | 1m45s | N/A |
| | HRank* | N/A | 10 | 69.758 | 59.336 | -10.422 | 1.77 **M** (15.14%) | 49m53s | N/A |
| | Random* | N/A | 11 | 69.758 | 57.102 | -12.656 | 3.40 **M** (29.10%) | 0.020s | N/A |
| | LAMP | N/A | 12 | 69.758 | 54.368 | -15.390 | 1.05 **M** (8.95%) | 0.030s | N/A |
| | BNScale | GroupLASSO | 1 | 69.758 | 63.729 | -6.029 | 6.77 **M** (57.91%) | 0.026s | 2h41m15s |
| | BNScale | BNScale | 2 | 69.758 | 63.671 | -6.087 | 6.98 **M** (59.71%) | 0.026s | 1h54m9s |
| | MagnitudeL2 | GroupNorm | 3 | 69.758 | 63.117 | -6.641 | 8.17 **M** (69.89%) | 0.038s | 3h4m27s |
| | MagnitudeL2 | GrowingReg | 4 | 69.758 | 63.042 | -6.716 | 9.04 **M** (77.33%) | 0.038s | 3h12m21s |
| | MagnitudeL2 | GroupLASSO | 5 | 69.758 | 62.814 | -6.944 | 9.27 **M** (83.54%) | 0.038s | 3h10m44s |
| 4x | FPGM | N/A | 1 | 69.758 | 61.442 | -8.316 | 6.98 **M** (59.68%) | 0.029s | N/A |
| | BNScale | N/A | 2 | 69.758 | 61.212 | -8.546 | 5.73 **M** (49.06%) | 0.026s | N/A |
| | MagnitudeL1 | N/A | 3 | 69.758 | 60.760 | -8.998 | 8.14 **M** (69.65%) | 0.023s | N/A |
| | MagnitudeL2 | N/A | 4 | 69.758 | 60.438 | -9.320 | 8.25 **M** (70.54%) | 0.038s | N/A |
| | Taylor* | N/A | 5 | 69.758 | 59.514 | -10.244 | 0.97 **M** (8.27%) | 22.487S | N/A |
| | ThiNet* | N/A | 6 | 69.758 | 57.228 | -12.53 | 0.61 **M** (5.26%) | 15.645s | N/A |
| | OBD-C* | N/A | 7 | 69.758 | 55.224 | -14.534 | 1.16 **M** (9.96%) | 24.096s | N/A |
| | HRank* | N/A | 8 | 69.758 | 53.398 | -16.360 | 0.99 **M** (8.48%) | 49m53s | N/A |
| | CP* | N/A | 9 | 69.758 | 52.602 | -17.156 | 0.40 **M** (3.45%) | 2m21s | N/A |
| | LAMP | N/A | 10 | 69.758 | 51.348 | -18.410 | 0.79 **M** (6.77%) | 0.030s | N/A |
| | Random* | N/A | 11 | 69.758 | 49.994 | -19.764 | 2.73 **M** (23.38%) | 0.020s | N/A |
| | OBD-Hessian* | N/A | 12 | 69.758 | 46.904 | -22.854 | 0.59 **M** (5.02%) | 1m45s | N/A |
| | MagnitudeL2 | GroupNorm | 1 | 69.758 | 61.106 | -8.652 | 7.77 **M** (66.47%) | 0.038s | 3h4m27s |
| | MagnitudeL2 | GroupLASSO | 2 | 69.758 | 60.771 | -8.987 | 8.12 **M** (69.46%) | 0.038s | 3h10m44s |
| | BNScale | GroupLASSO | 3 | 69.758 | 60.221 | -9.537 | 5.41 **M** (46.28%) | 0.026s | 2h41m15s |
| | MagnitudeL2 | GrowingReg | 4 | 69.758 | 60.127 | -9.631 | 8.31 **M** (71.09%) | 0.038s | 3h12m21s |
| | BNScale | BNScale | 5 | 69.758 | 60.043 | -9.715 | 5.32 **M** (45.51%) | 0.026s | 1h54m9s |

Table 19: Leaderboard of ResNet50 on ImageNet at three different speedup ratios. Global pruning with 10% group-wise protection is adapted.

| Speed Up | Method | | Rank | Base | Pruned | △ Acc | Pruning Ratio | Step Time | Reg Time |
|---|---|---|---|---|---|---|---|---|---|
| | Importance | Regularizer | | | | | | | |
| 2x | FPGM | N/A | 1 | 76.128 | 75.566 | -0.562 | 14.75 M (57.70%) | 0.538s | N/A |
| | OBD-C* | N/A | 2 | 76.128 | 74.361 | -1.767 | 12.94 M (50.65%) | 25.448s | N/A |
| | MagnitudeL1 | N/A | 3 | 76.128 | 74.118 | -2.01 | 18.17 M (71.09%) | 0.183s | N/A |
| | MagnitudeL2 | N/A | 4 | 76.128 | 73.684 | -2.444 | 18.26 M (71.44%) | 0.081s | N/A |
| | ThiNet* | N/A | 5 | 76.128 | 72.969 | -3.159 | 9.44 M (36.96%) | 43.444s | N/A |
| | Taylor* | N/A | 6 | 76.128 | 72.101 | -4.027 | 13.26 M (51.87%) | 25.590s | N/A |
| | OBD-Hessian* | N/A | 7 | 76.128 | 71.664 | -4.464 | 6.59 M (25.78%) | 6m9s | N/A |
| | BNScale | N/A | 8 | 76.128 | 71.812 | -4.316 | 17.29 M (67.66%) | 0.118s | N/A |
| | CP* | N/A | 9 | 76.128 | 71.410 | -4.718 | 4.75 M (18.57%) | 6m43s | N/A |
| | Random* | N/A | 10 | 76.128 | 71.399 | -4.729 | 12.86 M (50.30%) | 0.091s | N/A |
| | LAMP | N/A | 11 | 76.128 | 71.248 | -4.88 | 5.98 M (23.40%) | 0.103s | N/A |
| | HRank* | N/A | 12 | 76.128 | 69.865 | -6.263 | 9.53 M (37.27%) | 1h7m20s | N/A |
| | MagnitudeL2 | GroupLASSO | 1 | 76.128 | 73.661 | -2.467 | 17.68 M (69.16%) | 0.081s | 3h45m17s |
| | BNScale | BNScale | 2 | 76.128 | 73.343 | -2.785 | 17.68 M (69.17%) | 0.118s | 2h41m56s |
| | MagnitudeL2 | GroupNorm | 3 | 76.128 | 73.297 | -2.831 | 11.51 M (45.02%) | 0.081s | 3h43m21s |
| | BNScale | GroupLASSO | 4 | 76.128 | 73.176 | -2.952 | 17.43 M (68.21%) | 0.118s | 3h7m44s |
| | MagnitudeL2 | GrowingReg | 5 | 76.128 | 73.110 | -3.018 | 17.57 M (68.74%) | 0.081s | 3h51m10s |
| 3x | MagnitudeL1 | N/A | 1 | 76.128 | 73.774 | -2.354 | 15.42 M (60.34%) | 0.183s | N/A |
| | MagnitudeL2 | N/A | 2 | 76.128 | 73.542 | -2.856 | 14.37 M (56.23%) | 0.081s | N/A |
| | FPGM | N/A | 3 | 76.128 | 73.146 | -2.982 | 11.38 M (44.53%) | 0.538s | N/A |
| | Taylor* | N/A | 4 | 76.128 | 72.276 | -3.852 | 6.17 M (24.15%) | 25.590s | N/A |
| | OBD-C* | N/A | 5 | 76.128 | 72.702 | -3.426 | 7.74 M (30.29%) | 25.448s | N/A |
| | BNScale | N/A | 6 | 76.128 | 71.453 | -4.675 | 14.30 M (55.94%) | 0.118s | N/A |
| | ThiNet* | N/A | 7 | 76.128 | 70.994 | -5.134 | 4.77 M (18.68%) | 43.444s | N/A |
| | OBD-Hessian* | N/A | 8 | 76.128 | 69.476 | -6.652 | 3.38 M (13.24%) | 6m9s | N/A |
| | LAMP | N/A | 9 | 76.128 | 67.056 | -9.072 | 2.79 M (10.92%) | 0.103s | N/A |
| | HRank* | N/A | 10 | 76.128 | 66.134 | -9.994 | 6.65 M (26.02%) | 1h7m20s | N/A |
| | Random* | N/A | 11 | 76.128 | 65.314 | -10.814 | 8.91 M (34.87%) | 0.091s | N/A |
| | CP* | N/A | 12 | 76.128 | 64.536 | -11.592 | 1.90 M (7.43%) | 6m43s | N/A |
| | MagnitudeL2 | GroupLASSO | 1 | 76.128 | 71.811 | -4.317 | 14.01 M (54.81%) | 0.081s | 3h45m17s |
| | MagnitudeL2 | GroupNorm | 2 | 76.128 | 71.551 | -4.577 | 14.77 M (57.79%) | 0.081s | 3h43m21s |
| | BNScale | GroupLASSO | 3 | 76.128 | 71.507 | -4.621 | 14.25 M (55.75%) | 0.118s | 3h7m44s |
| | BNScale | BNScale | 4 | 76.128 | 71.399 | -4.729 | 14.81 M (57.94%) | 0.118s | 2h41m56s |
| | MagnitudeL2 | GrowingReg | 5 | 76.128 | 71.259 | -4.869 | 14.91 M (58.33%) | 0.081s | 3h51m10s |
| 4x | FPGM | N/A | 1 | 76.128 | 70.966 | -5.162 | 8.78 M (34.37%) | 0.538s | N/A |
| | MagnitudeL2 | N/A | 2 | 76.128 | 70.866 | -5.262 | 11.88 M (46.49%) | 0.081s | N/A |
| | MagnitudeL1 | N/A | 3 | 76.128 | 70.471 | -5.657 | 11.94 M (46.72%) | 0.183s | N/A |
| | OBD-C* | N/A | 4 | 76.128 | 70.156 | -5.972 | 6.00 M (23.48%) | 25.448s | N/A |
| | Taylor* | N/A | 5 | 76.128 | 69.063 | -7.065 | 3.48 M (13.63%) | 25.590s | N/A |
| | BNScale | N/A | 6 | 76.128 | 68.851 | -7.277 | 11.94 M (46.72%) | 0.118s | N/A |
| | ThiNet* | N/A | 7 | 76.128 | 68.468 | -7.660 | 2.79 M (10.91%) | 43.444s | N/A |
| | OBD-Hessian* | N/A | 8 | 76.128 | 65.106 | -11.022 | 2.93 M (11.45%) | 6m9s | N/A |
| | CP* | N/A | 9 | 76.128 | 64.754 | -11.374 | 1.36 M (5.33%) | 6m43s | N/A |
| | LAMP | N/A | 10 | 76.128 | 63.102 | -13.026 | 2.77 M (24.71%) | 0.103s | N/A |
| | HRank* | N/A | 11 | 76.128 | 62.964 | -13.164 | 4.39 M (17.16%) | 1h7m20s | N/A |
| | Random* | N/A | 12 | 76.128 | 61.244 | -14.884 | 6.73 M (26.33%) | 0.091s | N/A |
| | MagnitudeL2 | GroupLASSO | 1 | 76.128 | 69.897 | -6.231 | 12.17 M (47.61%) | 0.081 | 3h45m17s |
| | MagnitudeL2 | GroupNorm | 2 | 76.128 | 69.137 | -6.991 | 12.31 M (48.16%) | 0.081 | 3h43m21s |
| | BNScale | BNScale | 3 | 76.128 | 68.914 | -7.214 | 11.97 M (46.83%) | 0.118 | 2h41m56s |
| | MagnitudeL2 | GrowingReg | 4 | 76.128 | 68.759 | -7.369 | 11.94 M (46.71%) | 0.081 | 3h51m10s |
| | BNScale | GroupLASSO | 5 | 76.128 | 68.446 | -7.682 | 12.06 M (47.18%) | 0.118 | 3h7m44s |

Table 20: The leaderboard of ViT-small on ImageNet at three different speedup ratios. Global pruning with 10% group-wise protection is adapted.

| Speed Up | Method | | Rank | Base | Pruned | △ Acc | Parameters | Step Time | Reg Time |
|---|---|---|---|---|---|---|---|---|---|
| | Importance | Regularizer | | | | | | | |
| 2x | FPGM | N/A | 1 | 78.588 | 69.248 | -9.34 | 10.365 M (47.01%) | 0.937s | N/A |
| | Random* | N/A | 2 | 78.588 | 68.810 | -9.778 | 9.305 M (42.20%) | 0.888s | N/A |
| | LAMP | N/A | 3 | 78.588 | 68.724 | -9.864 | 10.169 M (46.12%) | 1.284s | N/A |
| | MagnitudeL1 | N/A | 4 | 78.588 | 68.602 | -9.986 | 10.375 M (47.05%) | 1.005s | N/A |
| | MagnitudeL2 | N/A | 5 | 78.588 | 68.316 | -10.272 | 10.346 M (46.92%) | 0.995s | N/A |
| | OBD-Hessian* | N/A | 6 | 78.588 | 67.514 | -11.074 | 10.334 M (46.87%) | 6m40s | N/A |
| | Taylor* | N/A | 7 | 78.588 | 67.400 | -11.188 | 10.468 M (47.47%) | 27.634s | N/A |
| | CP* | N/A | 7 | 78.588 | 67.400 | -11.188 | 10.334 M (46.87%) | 15m4s | N/A |
| | ThiNet* | N/A | 8 | 78.588 | 63.914 | -14.674 | 6.439 M (29.20%) | 3m17s | N/A |
| | MagnitudeL2 | GrowingReg | 1 | 78.588 | 68.715 | -9.873 | 10.359 M (46.98%) | 0.995s | 5h10m31s |
| | MagnitudeL2 | GroupNorm | 2 | 78.588 | 68.594 | -9.994 | 10.363 M (47.00%) | 0.995s | 5h21m21s |
| | MagnitudeL2 | GroupLASSO | 3 | 78.588 | 68.350 | -10.238 | 10.360 M (46.98%) | 0.995s | 5h15m13s |
| 3x | MagnitudeL1 | N/A | 1 | 78.588 | 63.120 | -15.468 | 6.57 M (29.79%) | 1.005s | N/A |
| | LAMP | N/A | 2 | 78.588 | 62.538 | -16.050 | 6.08 M (27.57%) | 1.284s | N/A |
| | MagnitudeL2 | N/A | 3 | 78.588 | 62.342 | -16.246 | 6.37 M (28.89%) | 0.995s | N/A |
| | Taylor* | N/A | 4 | 78.588 | 61.582 | -17.006 | 6.62 M (30.01%) | 27.634s | N/A |
| | FPGM | N/A | 5 | 78.588 | 60.660 | -17.928 | 5.701 M (25.85%) | 0.937s | N/A |
| | CP* | N/A | 6 | 78.588 | 56.626 | -21.962 | 6.778 M (30.74%) | 15m4s | N/A |
| | OBD-Hessian* | N/A | 7 | 78.588 | 54.796 | -23.792 | 6.39 M (28.98%) | 6m40s | N/A |
| | ThiNet* | N/A | 8 | 78.588 | 49.654 | -28.934 | 5.113 M (23.19%) | 3m17s | N/A |
| | Random* | N/A | 9 | 78.588 | 44.654 | -33.954 | 4.95 M (22.45%) | 0.888s | N/A |
| | MagnitudeL2 | GrowingReg | 1 | 78.588 | 62.608 | -15.980 | 6.57 M (29.81%) | 0.995s | 5h10m31s |
| | MagnitudeL2 | GroupNorm | 2 | 78.588 | 61.716 | -16.872 | 6.88 M (31.20%) | 0.995s | 5h21m21s |
| | MagnitudeL2 | GroupLASSO | 3 | 78.588 | 61.340 | -17.248 | 6.57 M (29.13%) | 0.995s | 5h15m13s |
| 4x | MagnitudeL1 | N/A | 1 | 78.588 | 59.950 | -18.638 | 5.06 M (22.93%) | 1.005s | N/A |
| | MagnitudeL2 | N/A | 2 | 78.588 | 59.082 | -19.506 | 4.89 M (22.15%) | 0.995s | N/A |
| | Taylor* | N/A | 3 | 78.588 | 57.650 | -20.938 | 4.80 M (21.76%) | 27.634s | N/A |
| | LAMP | N/A | 4 | 78.588 | 55.750 | -22.838 | 4.32 M (19.57%) | 1.284s | N/A |
| | FPGM | N/A | 5 | 78.588 | 48.258 | -30.33 | 3.25 M (14.74%) | 0.937 | N/A |
| | OBD-Hessian* | N/A | 6 | 78.588 | 36.600 | -41.988 | 4.25 M (19.27%) | 6m40s | N/A |
| | CP* | N/A | 7 | 78.588 | 42.574 | -36.014 | 5.253 M (23.82%) | 15m4s | N/A |
| | ThiNet* | N/A | 8 | 78.588 | 28.422 | -50.166 | 2.669 M (12.10%) | 3m17s | N/A |
| | Random* | N/A | 9 | 78.588 | 27.722 | -50.866 | 2.76 M (12.54%) | 0.888s | N/A |
| | MagnitudeL2 | GrowingReg | 1 | 78.588 | 59.630 | -18.958 | 4.56 M (20.66%) | 0.995s | 5h10m31s |
| | MagnitudeL2 | GroupLASSO | 2 | 78.588 | 57.312 | -21.276 | 4.59 M (20.81%) | 0.995s | 5h15m13s |
| | MagnitudeL2 | GroupNorm | 3 | 78.588 | 56.446 | -22.142 | 4.77 M (21.62%) | 0.995s | 5h21m21s |

