# OpenReview forum: "PruningBench: A Comprehensive Benchmark of Structural Pruning"
_ICLR.cc/2025/Conference — ICLR 2025 Conference Withdrawn Submission_

### Official Review · Reviewer_3ZNP · 2024-11-02

**Soundness:** 3
**Presentation:** 3
**Contribution:** 3
**Rating:** 8
**Confidence:** 3

**Summary:**

The paper proposed a unified and consistent framework, namely PruningBench, for evaluating the effectiveness of diverse structural pruning techniques. 16 structural pruning methods are systematically evaluated over classification or detection tasks with CNN based or ViT-samll architectures.

**Strengths:**

1. Extensive experiments, detailed experimental setups, and relevant analysis on observations are presented.
2. The proposed method greatly facilitates a fairer comparison of structured pruning methods.

**Weaknesses:**

1. It would be best to include more recent pruning methods from the last three years.
2. The experiments mainly focus on CNN-based architectures. For vit pruning, why were experiments not conducted on different scales such as ViT-Tiny and ViT-Base, or on different variants like DeiT and Swin? And the pruned Swin model for detection tasks?
3. The potential impact of the unified framework on the performance of pruning methods should be discussed further. In particular, whether the modifications made to adapt the methods to the framework can fairly and reasonably reflect the performance of the original methods.

**Questions:**

See weakness.

---

### Official Review · Reviewer_E16V · 2024-11-03

**Soundness:** 3
**Presentation:** 3
**Contribution:** 2
**Rating:** 5
**Confidence:** 4

**Summary:**

This paper introduces PruningBench, a benchmarking framework for vision-based model structural pruning. Specifically, the framework supports 16 different pruning methods that are tested over four classification and detection datasets and across five convolutional and ViT-based models. The evaluation groups results by speedup ratio and draws conclusions about the effect of architecture choice, local vs. global pruning, parameters vs. operations as well as efficiency and scalability of various methods, which yields valuable insights for ML practitioners and researchers.

**Strengths:**

* The paper attempts to tackle a well-motivated problem in the literature of structural pruning, that of inconsistencies in the evaluation methodology and baselines of various papers. The paper tries to fill this gap by standardizing the evaluation of new methods against a set of baselines over different dimensions and metrics.
* I appreciate the amount of effort that has been put into running these experiments across so many setups and baselines.
* I particularly liked the question-answering structure of the evaluation, and appreciate the insights shown in some behaviors.

**Weaknesses:**

* The paper is not as generic as it is posed to be, and some claims are not yet implemented. Specifically:
    - The paper claims to be a generic structural-based pruning benchmarking suite, but only shows results on vision tasks of classification and detection.
    - Code and public leaderboard infrastructure are not available at the time of submission.
    - Detection models are only YOLO-based.
* The paper has missed the potential of quantifying the performance gains of structural pruning on actual devices and showcasing real benefits of deployment.
* There are various typos and wording issues in the submissions that the authors should fix, especially over core terms of the paper.

**Questions:**

### Generality

* Can PruningBench support methods from the literature that sparsify the network over a low-rank [a] and/or dynamic width [b] representation of the weights?
* It would be worth expanding on the transformer (e.g. DeTR [c]) or SSM-based (e.g. VisionMamba [d]) architectures instead of integrating the now quite old by now VGG architecture.
* The selected models do not span across the very small (e.g. MobileNet/EfficientNet) or very large sizes (e.g. ViT-Large).
* How did the authors select the methods to implement in their framework vs. others (e.g. [e,f]) and how are they planning to expand their support for alternative methods from the literature.

### Contributions

* I found parameter grouping as a contribution to be somewhat orthogonal to the rest of the manuscript. Maybe the authors could treat this contribution as a showcase of the type of contribution PruningBench can enable.
* The Appendix has a lot of information that is worth referencing from the main manuscript.
* Are the authors planning to expand their support on other dense vision-related tasks, such as semantic segmentation?

### Evaluation

* The evaluation is missing an experimental setup version. This is needed, especially given the step and reg time quotes on Tables 2-3.
* Generally, the paper does a fair job at reporting results and asking interesting research questions, but I felt that the authors have not developed much insight about why these results manifest in many cases. Maybe uttering some hypotheses or putting such investigation as future work would enable further research in the field.
* It would be worth exploring the effects of pruning on actual device speedup and efficiency:
    - How do the pre-defined speedup ratios translate to latency or throughput performance gains on actual devices?
    - In the parameters vs. FLOPs, an interesting correlation with compute time and peak memory usage would be very useful. Also quantifying the energy gains would be a step towards sustainable AI.

Some additional question worth answering in the evaluation:

* How does the initial model size affect the redundancy in the final number of parameters. i.e. is it better to start with an overprovisioned model and depend on structural pruning for compression, or use a smaller model straightaway?
* Additionally, are there specific architectural choices (e.g. normalization layers, skip connections, weight-sharing etc.) affect the "pruning ability" of a model?
* How does pruning compare or can be combined with other compression methods [g].
* How do non-IID settings (e.g. in Federated Learning) affect the training and pruning ability of a model?


[a] Yu, J., Yang, L., Xu, N., Yang, J., & Huang, T. (2019). Slimmable neural networks. In 7th International Conference on Learning Representations, ICLR 2019.
[b] Horváth, S., Laskaridis, S., Rajput, S., & Wang, H. (2024). Maestro: Uncovering Low-Rank Structures via Trainable Decomposition. Forty-First International Conference on Machine Learning (ICML).
[c] Carion, N., Massa, F., Synnaeve, G., Usunier, N., Kirillov, A., & Zagoruyko, S. (2020, August). End-to-end object detection with transformers. In European conference on computer vision (pp. 213-229). Cham: Springer International Publishing.
[d] Zhu, L., Liao, B., Zhang, Q., Wang, X., Liu, W., & Wang, X. (2024). Vision mamba: Efficient visual representation learning with bidirectional state space model. Forty-First International Conference on Machine Learning (ICML).
[e] Chen, T., Liang, L., Tianyu, D. I. N. G., Zhu, Z., & Zharkov, I. (2023, January). OTOv2: Automatic, Generic, User-Friendly. In International Conference on Learning Representations.
[f] Chen, J., Chen, S., & Pan, S. J. (2020). Storage efficient and dynamic flexible runtime channel pruning via deep reinforcement learning. Advances in neural information processing systems, 33, 14747-14758.
[g] Kuzmin, A., Nagel, M., Van Baalen, M., Behboodi, A., & Blankevoort, T. (2024). Pruning vs quantization: which is better?. Advances in neural information processing systems, 36.

---

### Official Review · Reviewer_DKGM · 2024-11-04

**Soundness:** 2
**Presentation:** 2
**Contribution:** 2
**Rating:** 3
**Confidence:** 4

**Summary:**

The paper benchmarks 16 existing structural pruning methods on image classification and object detection models on CIFAR, ImageNet and COCO datasets.

**Strengths:**

Benchmarking structural pruning methods on the same set of data and models help address the inconsistency issue in performance evaluation of these methods. Certain insights from the benchmarking is consistent with observations from the literature, such as next-generation vision models are harder to compress.

**Weaknesses:**

1. Many work listed in table 1 and cited by this paper are from 2020 or earlier. Does the problem only exist in earlier pruning work? Why methods like CAP Kuznedelev et al. (2024) are not included in the benchmarking.

2. The paper argues that evaluation done in the comparison between the original and pruned models is limited. Model-specific compression has its own value, I am wondering why the comparison between a model and its pruned version is not enough in this context considering model compression is often used in deployment on specific environment.

3. DepGraph is used as a standard for weight grouping in the benchmarking. How to ensure that the results from DepGraph are correct and not misleading. How does this benchmark count for other grouping or weight correlation methods other than the one used in DepGraph?

Minor issue: line 203, duplicated references to Fang et al., 2023.

**Questions:**

See weaknesses.

---

### Official Review · Reviewer_ZWox · 2024-11-11

**Soundness:** 3
**Presentation:** 3
**Contribution:** 2
**Rating:** 3
**Confidence:** 4

**Summary:**

This paper presents PruningBench, a benchmark for evaluating and comparing structured pruning techniques. It observes that publications of individual pruning techniques suffer from flaws such as limited comparisons to SOTA, inconsistent experiment settings, and comparisons without controlling variables. It proposes a unified and consistent framework for evaluating such techniques, and instantiates it using 16 of them from the literature. They encompass different model architectures (CNNs and ViTs) and tasks (image classification and detection). Finally, it derives empirical observations such as the impact of model architectures, speedup ratio, and evaluation dataset choice on leaderboard rankings, as well as the computation costs and performance of the evaluated techniques.

**Strengths:**

1. Structured pruning is indeed becoming an increasingly important method to compress ever-larger models, and the paper makes a strong case for a consistent and unified framework to evaluate techniques in this field.

2. The paper instantiates the framework with a large number of techniques which demonstrates its generality. These include both sparsifying-stage methods and pruning-stage methods. Further, the latter include both data-free and data-driven methods.

3. The use of DepGraph to automatically group network parameters is proposed to avoid the labor effort and the group divergence by manually-designed grouping.  Furthermore, iterative pruning is proposed where a portion of parameters are removed per iteration until the controlled variable (e.g., FLOPS) is reached. This standardized framework ensures more equitable and comprehensible comparisons among various pruning methods.

**Weaknesses:**

1. The paper only evaluates vision models (CNNs and ViTs) on vision benchmarks (CIFAR and ImageNet). Pruning is far more generally applicable including to language models, where this kind of benchmarking framework would be even more valuable and, arguably, yield more interesting insights (see 2 below).

2. The findings (Q1-Q5) are not particularly interesting or surprising. I suspect a key reason for this is the limitation of the study to vision models. For instance, one of the findings is that no single method consistently outperforms the others across all settings and tasks.  I am left wondering whether the 16 chosen techniques are even popular in the vision community.

3. It would be useful to understand the qualitative impact on the drop in accuracy resulting from pruning. Only quantitative metrics are provided and it is not clear how to interpret them without a qualitative assessment.

**Questions:**

Please see Weaknesses.  I would especially like to know why pruning techniques for only vision models were considered as opposed to language models.

---

### Note · Authors · 2024-12-02

I have read and agree with the venue's withdrawal policy on behalf of myself and my co-authors.